# Lin−CCR2+ hematopoietic stem and progenitor cells overcome resistance to PD-1 blockade

Catherine T. Flores [1], Tyler J. Wildes [1], Jeffrey A. Drake[1], Ginger L. Moore[1], Bayli DiVita Dean[1], Rebecca S. Abraham[1] & Duane A. Mitchell[1]

Immune checkpoint blockade using anti-PD-1 monoclonal antibodies has shown considerable promise in the treatment of solid tumors, but brain tumors remain notoriously refractory to treatment. In CNS malignancies that are completely resistant to PD-1 blockade, we found that bone marrow-derived, lineage-negative hematopoietic stem and progenitor cells (HSCs) that express C–C chemokine receptor type 2 (CCR2+) reverses treatment resistance and sensitizes mice to curative immunotherapy. HSC transfer with PD-1 blockade increases T-cell frequency and activation within tumors in preclinical models of glioblastoma and medulloblastoma. CCR2+HSCs preferentially migrate to intracranial brain tumors and differentiate into antigen-presenting cells within the tumor microenvironment and cross-present tumor-derived antigens to CD8+ T cells. HSC transfer also rescues tumor resistance to adoptive cellular therapy in medulloblastoma and glioblastoma. Our studies demonstrate a novel role for CCR2+HSCs in overcoming brain tumor resistance to PD-1 checkpoint blockade and adoptive cellular therapy in multiple invasive brain tumor models.

[1] University of Florida Brain Tumor Immunotherapy Program, Preston A. Wells, Jr. Center for Brain Tumor Therapy, Lillian S. Wells Department of Neurosurgery, University of Florida, 1149S Newell Dr, L2-100, Gainesville, FL 32611, USA. Correspondence and requests for materials should be addressed to C.T.F. (email: catherine.flores@neurosurgery.ufl.edu) or to D.A.M. (email: duane.mitchell@neurosurgery.ufl.edu)

mmunotherapy has emerged as a remarkably effective treatment modality, leading to clinical responses in both human and murine systems. The excitement around the two major modalities, immune checkpoint inhibitors and adoptive cellular therapy, is centered on their potentially broad clinical applicability across multiple cancers. Despite successes in the treatment of some advanced malignancies using cancer immunotherapy, the majority of patients with solid tumors demonstrate resistance to immune checkpoint blockade and adoptive cellular therapy[1–3]. Brain tumors have been notoriously difficult to treat using existing immunotherapeutic strategies[3]. In fact, a recent phase III trial failed to demonstrate survival benefit with PD-1 monotherapy against recurrent glioblastoma, an almost universally fatal brain tumor[3]. In addition, we have demonstrated in preclinical models that brain tumors differ in responsiveness to checkpoint inhibition, specifically to anti-PD-1[4]. Notwithstanding these results, the curative potential of immunotherapy is so great that understanding and overcoming treatment resistance is paramount. We have discovered a novel method of overcoming treatment resistance to both PD-1 and adoptive cellular therapy by employing a concomitant hematopoietic stem and progenitor cell (HSC) transfer.

Our previous work has demonstrated that the administration of bone marrow-derived HSCs is required to observe efficacy of adoptive cellular therapy against glioma in a preclinical model[5,6]. HSCs lead to significant accumulation of adoptively transferred tumor-reactive T cells within the tumor microenvironment[5,6]. Preclinical studies demonstrate that increasing activated anti-tumor T cells within the tumor microenvironment is an essential component for the immunologic rejection of tumors after either anti-PD-1 immune checkpoint inhibition or adoptive cellular therapy[2,7–10]. Recent elegant work has demonstrated that tumor-associated dendritic cells (DCs) within the tumor microenvironment play a major role in this accumulation of activated T cells in the context of both checkpoint blockade and adoptive cellular therapy[7,8]. This mechanism is so impactful that it has been strongly suggested that the absence of DCs in the tumor may possibly be a mechanism of treatment resistance to immunotherapy[7,8]. Here, we demonstrate that a subset of lineage negative (lin−) HSCs that express chemokine receptor type 2 (CCR2), herein referred to as CCR2+HSCs, have the capacity to migrate to intracranial tumors and differentiate into professional antigen-presenting cells (APCs) within the tumor microenvironment. This leads to increased intra-tumor T-cell activation after treatment with either PD-1 inhibition or adoptive cellular therapy. We demonstrate that combining CCR2+HSCs with immunotherapy leads to overcoming treatment resistance to monotherapeutic strategies.

We found that combinatorial CCR2+HSCs plus anti-PD-1 leads to increased median survival and long-term survivors in preclinical brain tumor models (glioblastoma and medulloblastoma) that are completely refractory to PD-1 treatment alone. Combination of CCR2+HSCs with adoptive cellular therapy also significantly extends survival in brain tumor-bearing mice. In addition, co-transfer of CCR2+HSCs with adoptive cellular therapy leads to the persistent activation status of adoptively transferred tumor-reactive T cells. We found that intravenously administered CCR2+HSCs migrate preferentially to the CNS tumor microenvironment, differentiate into CD11c+ APCs at the tumor site, and reprogram gene expression within the immunosuppressive tumor microenvironment, while targeting multiple suppressive pathways at once. Additionally, the APCs derived from CCR2+HSCs uniquely cross-present tumor-derived antigens to both endogenous and adoptively transferred T lymphocytes, leading to prolonged T-cell activation within brain tumors and enhanced tumor rejection. These studies demonstrate a unique role for CCR2+HSCs in overcoming brain tumor resistance to PD-1 blockade and adoptive cellular therapy.

## Results

**HSC transfer overcomes resistance to anti-PD-1 monotherapy.** We have explored treatment of syngeneic murine intracranial glioblastoma (KR158B) and a molecular subtype sonic hedgehog medulloblastoma (Ptc)[4,11] with monoclonal anti-PD-1 therapy (PD-1) and found both tumors to be completely refractory to immune checkpoint blockade with PD-1 monotherapy (Fig. 1a, b). Both these brain tumors express PD-L-1 on their cell surface in vivo yet are completely refractory to monotherapy[4].

Our prior publication demonstrated that the co-administration of HSCs with adoptive cellular therapy leads to increased recruitment of T cells into the tumor microenvironment and enhances the efficacy of adoptive cellular therapy targeting malignant gliomas[5,6]. We therefore explored the potential enhancing effects of HSCs with PD-1 blockade. The HSCs used in this study were isolated from fresh bone marrow using a magnetic bead lineage depletion kit (Miltenyi Biotec). The resulting lineage negative population is depleted of CD5, CD45R, CD11b, Gr-1, 7–4, and Ter-119.

KR158B glioma-bearing mice and administered either no treatment, HSCs only, PD-1 only, or HSC + PD-1 (Fig. 1a). This was repeated in mice with intracranial Ptc medulloblastoma (Fig. 1b). Therapeutic interventions were administered 5 days post intracranial tumor implantation, and in glioma-bearing mice, combinatorial HSC + PD-1 led to significant survival benefit over KR158B tumor only group ($p = 0.0005$), and over PD-1 only group ($p = 0.0005$). In mice with cerebellar Ptc medulloblastoma, combination HSC + PD-1 also led to a significant increase in median survival relative to both Ptc tumor only ($p = 0.0085$), and PD-1 only ($p = 0.0006$). This finding may have impactful implications for the use of immune checkpoint inhibitors against CNS malignancies.

**HSC transfer increases T-cell activation within tumor.** We then sought to determine the impact of HSC transfer on endogenous T cells with PD-1 blockade. To determine the effect of HSC co-transfer on the relative amounts of activated tumor infiltrating lymphocytes, we employed GREAT mice which have an interferon-gamma (IFNγ) promoter with IRES-eYFP reporter (Jackson Laboratories) to enable longitudinal evaluation of T-cell activation within the tumor microenvironment. Here, GREAT mice were implanted intracranially with KR158B glioma in the cerebral cortex. These tumor-bearing mice then received either no treatment, HSCs alone, PD-1 alone, or combination HSC + PD-1. Three weeks after HSC transfer, tumors and draining cervical lymph nodes were excised and analyzed for YFP+ expression by CD3+ cells as an indication of activation and IFNγ secretion of T cells in the tumor microenvironment (Fig. 1c) and tumor draining lymph nodes (Fig. 1d). Mice that received combinatorial HSC + PD-1 had significantly increased YFP+CD3+ cells within the tumor ($p = 0.0079$) (Fig. 1c) and draining lymph nodes (Fig. 1d) relative to groups that received PD-1 alone ($p = 0.0079$). To substantiate this observation of increased T-cell activation, we conducted genetic analysis using a real time PCR array, RT2 Profiler Array Cancer Inflammation and Immunity Crosstalk (Qiagen), on the tumors of mice that received either no treatment, HSCs only, PD-1 only, or HSC + PD-1. Analysis revealed that tumors from mice that received combinatorial HSC + PD-1 had >2-fold increase in gene expression of T-cell activation markers IFN-γ, FasL, Stat1, and TNFα over the group that received anti-PD-1 alone. Of significance, the group that received HSC transfer alone, relative to the no treatment group, showed >2-fold decrease in expression of markers associated with tumor-derived immunosuppression (iNOS, IDO, and TGF-β)

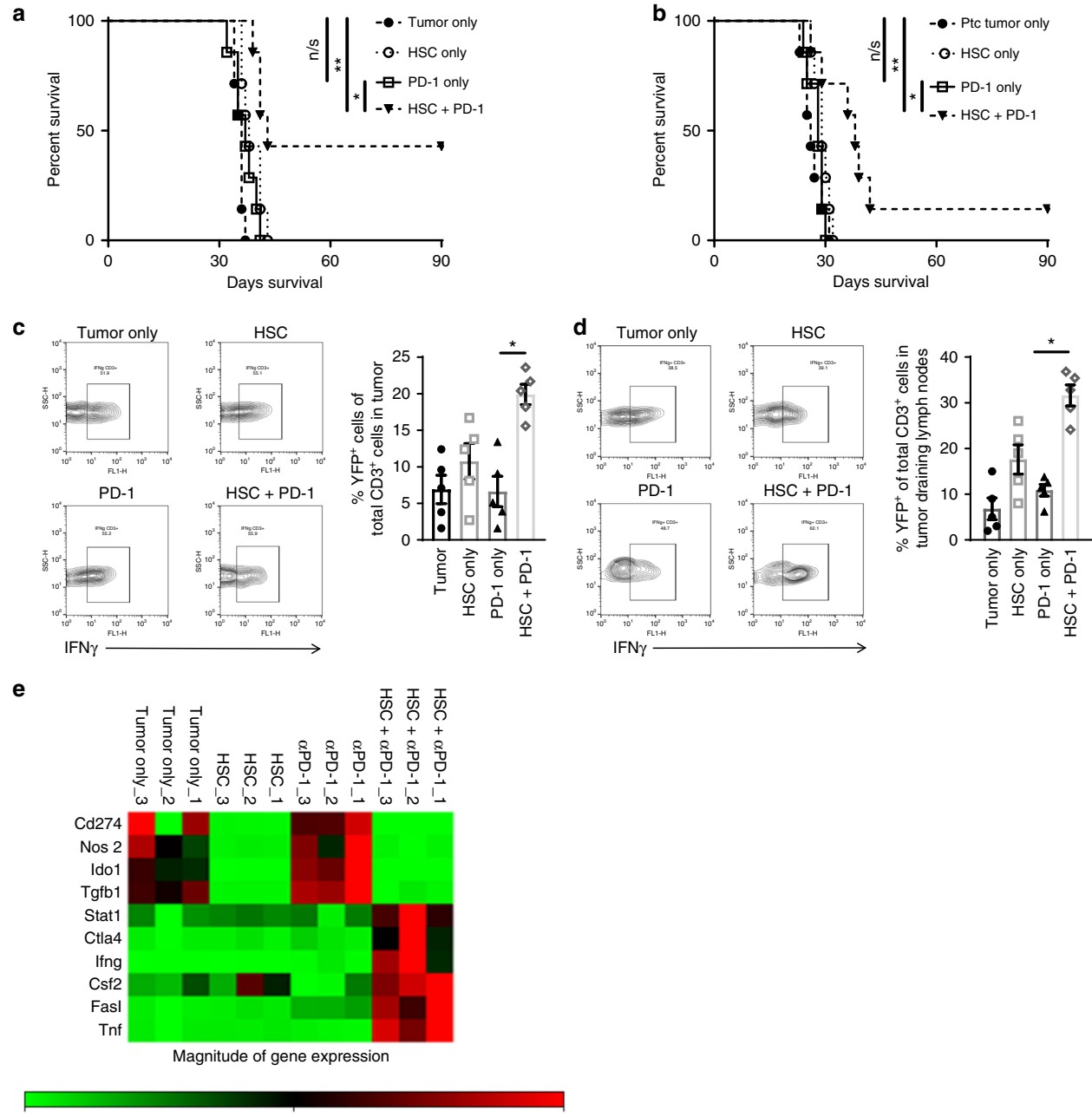

**Fig. 1** HSC co-transfer increases overall survival in brain tumors refractory to PD-1. **a** Intracranial KR158B glioma was treated with either no treatment, HSCs only, PD-1 only, or the combination HSC + PD-1 ($n = 7$ mice/group; Gehan–Breslow–Wilcoxon statistical test; *n/s $p = 0.3000$, *$p = 0.0024$ **$p = 0.0005$). **b** Cerebellar Ptc medulloblastoma was also treated with either no treatment, HSCs only, PD-1 only, HSC + PD-1 ($n = 7$ mice/group; Gehan–Breslow–Wilcoxon statistical test) *n/s $p = 0.476$, *$p = 0.0006$, **$p = 0.0085$. **c** GREAT mice with established KR158B tumors received either no treatment, HSCs only, PD-1 only, or HSC + PD-1. Tumors were excised and analyzed for relative expression of YFP using flow cytometry to compare relative YFP+ CD3+ expression in mice that received HSC + PD-1 relative to PD-1 only (*$p = 0.0079$, Mann–Whitney test; $n = 5$ mice/group). **d** GREAT mice with established KR158B tumors received either no treatment, HSCs only, PD-1 only, or HSC + PD-1. Tumor draining lymph nodes were excised and analyzed for relative expression of YFP using flow cytometry to compare relative YFP+ CD3+ expression in mice that received HSC + PD-1 only relative to PD-1 only (*$p = 0.0079$, Mann–Whitney test; $n = 5$ mice/group). **e** RNA was extracted from the same tumors as in panel C and RT2 PCR Array for Cancer Inflammation and Immunity Crosstalk was performed ($n = 3$ tumors/group). This figure is representative of relative expression of genes associated with T-cell activation and immunosuppressive tumor microenvironments that express >2-fold difference relative to housekeeping genes. All error bars represent s.e.m.

demonstrating a reprogramming of the global tumor microenvironment by HSC transfer (Fig. 1e).

**Route of HSC administration**. In these studies, HSCs were administered intravenously because in the clinical setting, blood and marrow stem and progenitor cell transfers are typically delivered via intravenous infusion. However, the biological activity described in these studies have specifically focused on the interactions within the brain tumor microenvironment, therefore we conducted experiments to determine if direct administration

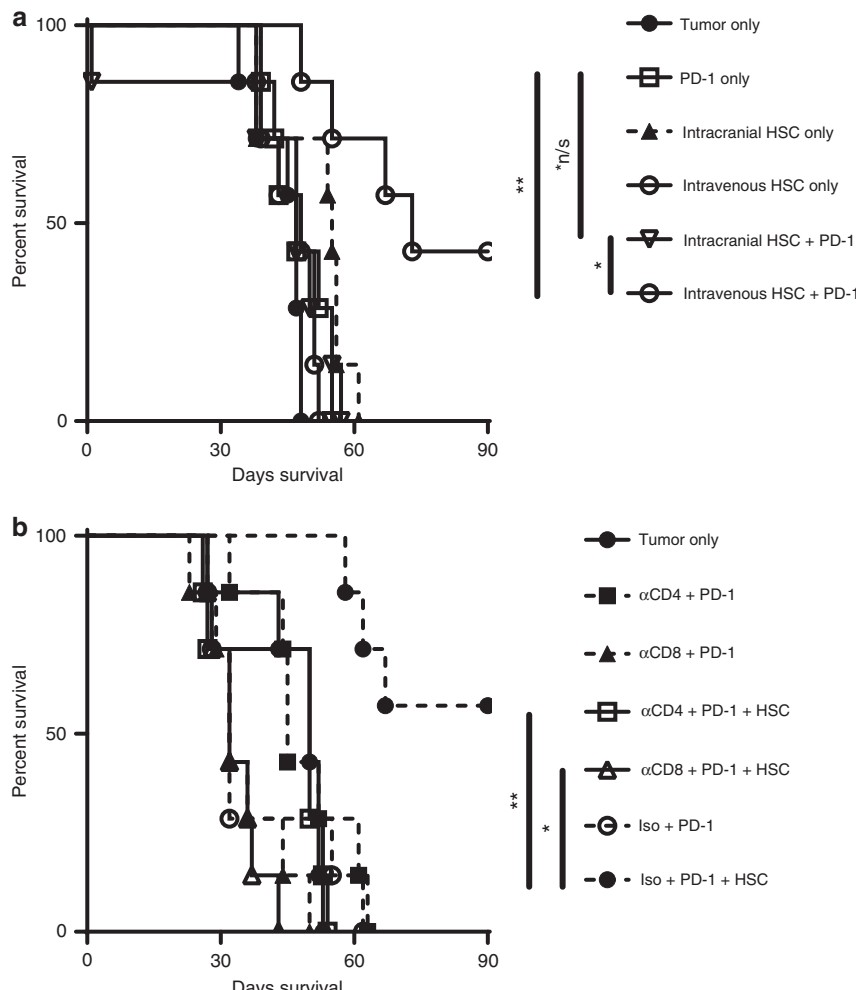

**Fig. 2** Efficacy of HSC + PD-1 is dependent on both CD4 and CD8 lymphocytes. **a** KR158B glioma tumor-bearing mice received no treatment, PD-1 only, intracranial HSCs, intravenous HSCs, intracranial HSC + PD-1, or intravenous HSCs + PD-1 and observed for survival (*$p = 0.0039$, **$p = 0.0056$, *n/s $p = 0.9491$, $n = 7$ mice/group; Gehan–Breslow–Wilcoxon statistical test). **b** Glioma-bearing mice were depleted of CD4[+] or CD8[+] cells, then received either PD-1 or HSC + PD-1. Isotype + HSC + PD-1 served as the positive control (*$p = 0.0001$, **$p = 0.0002$, $n = 7$ mice/group; Gehan–Breslow–Wilcoxon statistical test). All error bars represent s.e.m.

of HSCs into the tumor bed using stereotactic injection impacts efficacy. C57BL/6 mice were stereotactically implanted with KR158B glioma. Five days later, equal numbers of HSCs ($10^5$ HSCs) were administered either directly into the tumor bed via stereotactic injection, or tail vein injection. Tumor-bearing mice received either no treatment, PD-1 only, intracranial HSCs only, intravenous HSCs only, intracranial HSC + PD-1, or intravenous HSC + PD-1. Groups were then followed for survival (Fig. 2a). Administering HSCs directly into the tumor did not provide any survival benefit when combined with PD-1 relative to the no treatment group ($p = 0.9491$). The group that received intravenous HSCs + PD-1 had a significant increase in median survival over the intracranial HSC + PD-1 group ($p = 0.0039$) indicating that the route of administration of the cells is crucial to efficacy. These observations also indicate that either the biological activity of cells derived from the transferred HSCs requires interactions in the periphery and/or the intravenous route of delivery more effectively delivers HSCs to sites of invasive tumor growth than inoculation in situ into the tumor bed.

**CD4 and CD8 cells are necessary for efficacy**. T lymphocytes play a crucial role in the immunological rejection of solid tumors.

Here we sought to determine if CD4 or CD8 lymphocytes are necessary in mediating tumor rejection by HSC + PD-1 therapy. We first depleted endogenous CD4 or CD8 cells using depleting antibodies (αCD4 or αCD8) from C57BL/6 mice. Mice then received KR158B glioma followed with either no treatment, αCD4 + PD-1, αCD8 + PD-1, αCD4 + PD-1 + HSC, αCD8 + PD-1 + HSC, Isotype + PD-1, Isotype + PD-1 + HSC (both αCD4 and αCD8 depleting antibodies are the same isotype) (Fig. 2b). In all cohorts where either CD4 or CD8 were depleted, there was no survival benefit over the no treatment group. Both CD4 and CD8 cells are essential in the rejection of brain tumors by combinatorial HSC + PD-1.

**CCR2[+]HSCs + PD-1 increases T-cell activation within tumor**. The HSCs from the bone marrow is a heterogeneous population of stem and progenitor cells, thus, we sought to identify the subpopulation responsible for the observed increase in intratumor T-cell activation. HSCs were harvested and isolated into previously characterized stem and progenitor cell populations, Sca1[+]Lin[−]cells, cKit[+]Lin[−] cells, CD133[+]Lin[−] cells, CD38[+]Lin[−] cells, or CCR2[+]Lin[−] cells. We then employed our previously established method of generating tumor-specific T-cells ex vivo

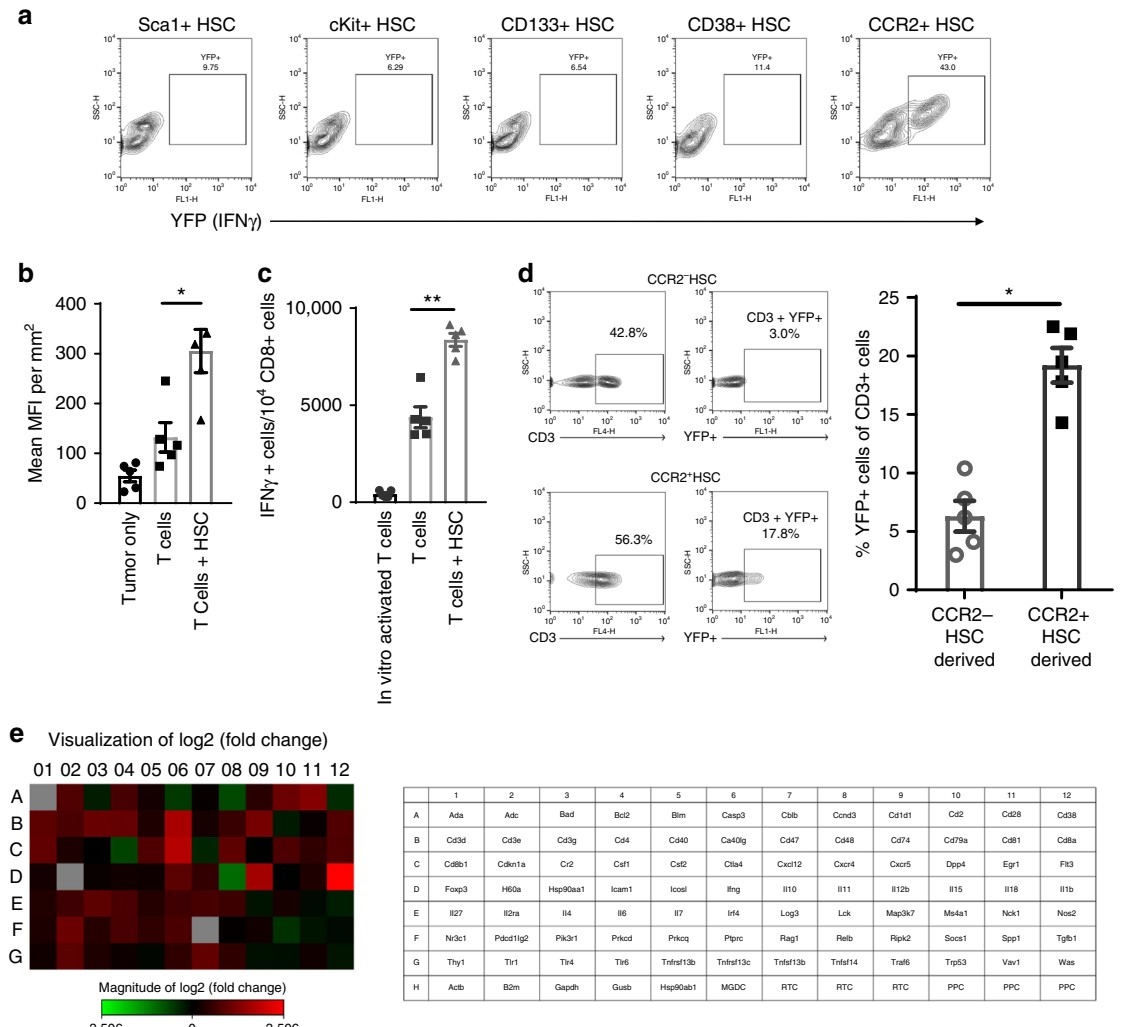

**Fig. 3** CCR2+HSCs increase T-cell activation within tumor after adoptive cell therapy. **a** Lineage negative HSCs were further isolated into stem and progenitor cell populations: Sca1 + HSCs, cKit + HSCs, CD133 + HSCs, CD38 + HSCs, CCR2+HSCs. Each progenitor cell population was transferred with tumor-reactive T-cells (generated from GREAT mice) into tumor-bearing mice. Relative amounts of YFP+ was measured in the tumors. **b** Late-stage tumor-bearing mice received adoptive T-cell therapy and dendritic cell (DC) vaccine with or without HSC transfer. Twenty-one days post transfer, brains were excised and sliced into sections, then analyzed for YFP+ cells within the tumor microenvironment by measuring relative MFI across brain sections (*$p =$ 0.0110, $n = 5$/group, Mann–Whitney test). **c** Late-stage tumor-bearing mice received adoptive T-cell therapy and DC vaccine with or without HSC transfer. Draining lymph nodes were harvested and YFP+ CD3+ cells were enumerated to determine the relative amounts of activated T cells, *$p =$ 0.0002, $n = 5$/ group, Mann–Whitney test). **d** Two groups of KR158 glioma-bearing mice received adoptive T-cell therapy with concomitant transfer of either CCR2−HSCs or CCR2+HSCs. Tumors were excised and relative amounts of YFP+ of CD3+ cells were measured with flow cytometry (*$p =$ 0.0002, $n = 5$/group, unpaired $t$-test). **e** Tumor bearing mice were treated with either CCR2+HSC + PD-1 or CCR2−HSC + PD-1. Tumors were excised and RNA was isolated to conduct RT2 PCR Array for T- and B-cell activation (Qiagen). Genes represented demonstrated >2-fold change in expression (this has been replicated twice). All error bars represent s.e.m.

where bone marrow-derived dendritic cells are pulsed with total tumor RNA and co-cultured with splenocytes of previously immunized mice[5,6]. These tumor-specific T-cells were generated using splenocytes of GREAT mice and were co-transferred into tumor-bearing mice that received infusion of different stem and progenitor cell populations (Sca1+HSCs, cKit+HSCs, CD133 +HSCs, CD38+HSCs, or CCR2+HSCs). Seven days post transfer, tumors were excised and processed into single cell suspensions. Using flow cytometry, relative frequencies of YFP+CD3+ cells from the milieu was measured. The group that received CCR2 +HSCs HSCs expressed the highest amount of CD3+YFP+ cells within the tumor microenvironment where an average of 42% ± 6.84% of CD3+ cells were found to be YFP+, more than the groups that received any of the other progenitor cell populations (Fig. 3a).

**CCR2+HSCs increase activation of tumor-specific T cells**. We previously demonstrated that HSCs co-transfer with adoptive cellular therapy mediates migration and engraftment of tumor-specific T cells as well as suppression of tumor growth in KR158B glioma[5]. We found that concomitant transfer of HSCs with tumor-specific T-cells significantly increases tumor-specific T-cell activation and IFNγ secretion within the tumor microenvironment (Fig. 3b) ($p =$ 0.0110), and in draining lymph nodes relative to groups that received T-cells alone (Fig. 3c) ($p =$ 0.002).

To determine the impact of CCR2+HSCs on T-cell activation within the tumor microenvironment, mice with established intracranial gliomas received adoptive transfer of tumor-specific T cells generated from GREAT mice. After T-cell transfer, mice then received intravenous administration of either CCR2+HSCs or CCR2−HSCs. One week post T-cell transfer, tumors were

excised and relative expression of YFP$^+$ on CD3$^+$ cells within the tumor was determined. The group that received CCR2$^+$HSCs expressed significantly more YFP$^+$CD3$^+$ cells than the group that received CCR2$^-$HSCs (19.2% vs 6.3%, $p = 0.0002$) (Fig. 3d). Genetic confirmation of increased T-cell activation was then conducted. RNA was isolated from tumors of mice that received adoptive transfer of tumor-specific T-cells concomitantly with either CCR2$^+$HSCs or CCR2$^-$HSCs. RT2 PCR Array for T- and B-cell activation was conducted, comparing relative gene expression between the two groups (Fig. 3e). Tumors from mice that received CCR2$^+$HSCs had >2-fold expression of genes associated with increased activated T-cells including CD3, CD8, IFN$\gamma$, and CD40L (Fig. 3e).

**CCR2$^+$HSCs migrate to intracranial tumor**. We sought to determine the kinetics of CCR2$^+$HSCs migration to intracranial tumor relative to CCR2$^-$HSCs. CCR2$^+$HSCs were isolated from bone marrow of DsRed transgenic mice (Jackson Laboratories), and CCR2$^-$HSCs were isolated from the bone marrow of GFP transgenic mice (Jackson Laboratories). DsRed$^+$CCR2$^+$HSCs and GFP$^+$CCR2$^-$HSCs were intravenously injected in equal amounts ($5 \times 10^5$ cells) into tumor-bearing mice that received lymphodepletion with 5 Gy total body irradiation. Twenty-four hours, 7 days, or 30 days later, mice were euthanized and their tumors were excised and processed into single cell suspensions for flow cytometry. Relative amounts of DsRed$^+$CCR2$^+$HSCs vs. GFP$^+$CCR2$^-$HSCs were measured within the tumor. Significantly more DsRed$^+$CCR2$^+$HSCs accumulated within intracranial tumor than GFP$^+$CCR2$^-$HSCs at 24 h ($p = 0.0010$) (Fig. 4a). By 7 days post transfer, however, there was significantly higher accumulation of cells derived from GFP$^+$CCR2$^-$HSCs relative to DsRed$^+$CCR2$^+$HSCs ($p = 0.0013$) and remained the same relative frequencies at 30 days post transfer ($p = 0.0003$) (Fig. 4a). Given the observed correlation between the transfer of CCR2$^+$HSCs and T-cell activation within the tumor microenvironment, we next conducted experiments to determine the phenotype and function of cells derived from CCR2$^+$HSCs within the tumor microenvironment.

**CCR2$^+$HSCs differentiate into dendritic cells**. CCR2 has been characterized to be expressed by monocyte precursor cells and is required for their entry into the CNS[12,13]. This knowledge, coupled with our observations of increased T-cell activation within brain tumors, led us to hypothesize that CCR2$^+$HSC-derived cells found within the brain tumor are differentiating into antigen-presenting cells, thereby activating lymphocytes within the brain tumor microenvironment. KR158B intracranial bearing mice received intravenous injection of either CCR2$^+$ or CCR2$^-$ HSCs that were isolated from bone marrow of GFP$^+$ mice. One week later, mice were euthanized and tumors were excised and processed into single cell suspensions. Phenotype analysis determined that once the cells derived from CCR2$^+$HSCs reach intracranial tumor after intravenous injection, they upregulate markers associated with antigen presenting cells, including CD11c, MHC-II, and CD80 (Fig. 4b). It is important to note that one week after cell transfer, cells originally derived from CCR2$^+$HSC downregulated CCR2 expression and have a very low frequency of either F4/80 or Ly6G/6C$^+$ cells, indicating the lack of myeloid-derived suppressor cells (MDSC) phenotype (Fig. 4c). In contrast, cells derived from CCR2$^-$HSCs upregulated markers associated with MDSCs including F4/80, Ly6G/6C and CCR2 (Fig. 4d). To determine the relative amount of CCR2$^+$HSC-derived cells vs. CCR2$^-$HSC-derived cells that upregulate a suppressive phenotype, we injected $5 \times 10^5$ of either DsRed$^+$CCR2$^+$HSC or GFP$^+$CCR2$^-$HSC into tumor-bearing mice.

Seven days post transfer, brain tumors were harvested and HSC-derived cells were isolated and phenotyped for F4/80, Ly6G/6C, and CCR2. We found that relative to CCR2$^+$HSC-derived cells, once in the brain tumor, cells that arose from CCR2$^-$HSCs expressed significantly more CCR2 ($p = 0.0035$), F4/80 ($p = 0.0138$), and Ly6G/6C ($p = 0.0001$) (Supplementary Fig. 1). This is a strong indicator that both cell populations have different functions within the tumor.

We next sought to determine if CCR2$^+$HSCs have the capacity to differentiate into functional dendritic cells. CCR2$^+$HSCs and CCR2$^-$HSCs were isolated from bone marrow of mice and cultured in vitro under previously published dendritic cell generation conditions which include GM-CSF and IL-4[5,14,15]. Cells originally derived from the CCR2$^+$HSCs expressed a distinct dendritic cell phenotype including CD11C, MHC-II, CD80, and CD86. Cells derived from CCR2$^-$HSCs express a similar phenotype but have notably high expression of Gr-1, indicating a suppressive phenotype (Supplementary Fig. 2).

The cells that arose from both CCR2$^+$HSCs and CCR2$^-$HSCs after culturing in dendritic cell media were pulsed with tumor-RNA. These were then used as targets for tumor-reactive T cells in vitro to determine if cognate tumor antigens were being presented by these cells as an indication of their antigen presenting cell function. KR158B tumor cells were used as a positive target control, and supernatant IFN$\gamma$ was then measured (Fig. 4e). T-cells secreted equal amounts of IFN$\gamma$ when used as effectors against either the tumor-RNA-pulsed dendritic cells derived from CCR2$^+$HSCs, or KR158B tumor cells themselves ($p = 0.5176$), demonstrating that CCR2$^+$HSCs-derived cells have the capacity to present antigen on their MHC. The resulting dendritic cells derived from CCR2$^+$HSCs were markedly superior in presenting tumor antigens in vitro over cells that arose from CCR2$^-$HSCs (Fig. 4e).

**CCR2$^+$HSC-derived cells cross-prime endogenous T cells**. To determine if HSC-derived cells found in the tumor extravasate from intracranial tumor to draining lymph nodes, DsRed$^+$HSCs, DsRed$^+$CCR2$^-$HSCs, or DsRed$^+$CCR2$^+$HSCs were directly injected into established intracranial KR158B tumors in vivo. After one week, draining cervical lymph nodes were harvested and analyzed for presence of DsRed$^+$ cells (Fig. 5a). The group that received CCR2$^+$HSCs had significantly more DsRed$^+$ cells in the lymph nodes over unsorted HSC group ($p = 0.0069$), and the group that received CCR2$^-$HSCs ($p = 0.021$) (Fig. 5a). To determine if the cells derived from CCR2$^+$HSCs that extravasate from intracranial tumor to the draining lymph nodes have the capacity to cross-prime endogenous T cells, tumor-bearing mice received intra-tumor injection of either DsRed$^+$HSCs, DsRed$^+$CCR2$^-$HSCs, or DsRed$^+$CCR2$^+$HSCs and treated with PD-1. Three weeks post-cell transfer, the tumor draining cervical lymph nodes were harvested and T cells were isolated using a T-cell magnetic bead depletion kit (Miltenyi Biotec). T cells were tested for anti-tumor function by co-culturing them against target KR158B tumor cells in vitro (Fig. 5b). Mice that received intra-cranial injection of CCR2$^+$HSCs demonstrated increased IFN-$\gamma$ secretion by peripheral T cells in response to tumor antigens, suggesting that these T cells were cross-primed, presumably by the cells derived from the CCR2$^+$HSCs that differentiated into antigen presenting cells in the tumor and extravasated into the draining lymph node (Fig. 5b). This experiment was repeated in our cerebellar Ptc medulloblastoma model (Fig. 5c). The cohort that received intratumor CCR2$^+$HSCs had endogenous T cells isolated from tumor draining lymph nodes that mounted the capacity to recognize Ptc tumor cells, secreting IFN$\gamma$ upon co-culture against Ptc tumor cell targets (Fig. 5c).

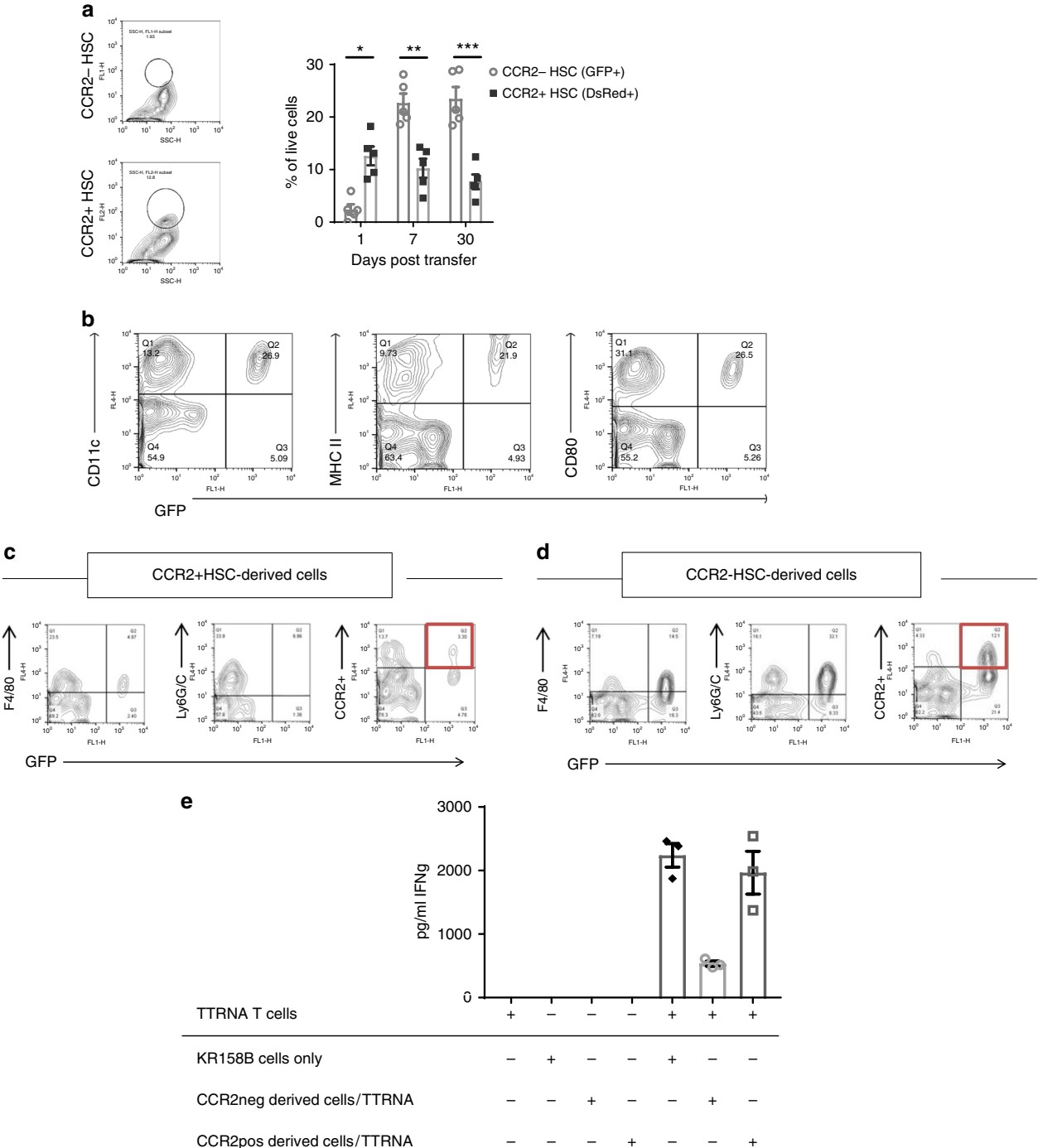

**Fig. 4** CCR2$^+$HSCs migrate to intracranial tumor and differentiate into dendritic cells. **a** Equal numbers of CCR2$^-$HSCs (isolated from GFP mice) and CCR2$^+$HSCs (isolated from DsRed mice) were injected into tumor-bearing mice. Relative amounts of cells derived from either GFP$^+$CCR2$^-$HSCs and DsRed$^+$CCR2$^+$HSCs were compared at 1 day (*$p = 0.0010$), 7 days (**$p = 0.0013$), and 30 days (***$p = 0.0003$) post transfer (Mann–Whitney tests; $n = 7$ mice per group). **b** In a separate experiment, CCR2$^+$HSCs were isolated from GFP$^+$ mice and intravenously co-transferred into tumor-bearing mice that received adoptive cell therapy with tumor-reactive T-cells. Twenty-one days post transfer, GFP$^+$ cells found in the tumor were phenotyped for makers of dendritic cells. **c**, **d** Both CCR2$^-$HSCs and CCR2$^+$HSCs were isolated from GFP$^+$ mice and co-transferred into tumor-bearing mice that received adoptive cell therapy with tumor-reactive T-cells. Twenty-one days post transfer, GFP$^+$ cells found in the tumors were analyzed for markers of a suppressive phenotype. **e** CCR2$^-$HSCs and CCR2$^+$HSCs were isolated and cultured in vitro in dendritic cell media containing GM-CSF and IL-4. Resulting cells were electroporated with total tumor RNA and used as targets for tumor-reactive T-cells. Supernatant IFNγ was measured (*$p = 0.0135$, *n/s $p = 0.5176$, $n = 5$ replicates, unpaired $t$-test). All error bars represent s.e.m.

**CCR2$^+$HSC-derived DCs cross-present to tumor-specific T cells.** Since the above experiment describes that CCR2$^+$HSC-derived cells can differentiate into functional APCs in vitro, we sought to determine if HSC-derived cells can cross-prime adoptively transferred tumor-reactive T cells used for adoptive cellular therapy. We first determined the capacity of the bulk HSC population to differentiate into antigen presenting cells within the tumor microenvironment. Here, intracranial KR158B

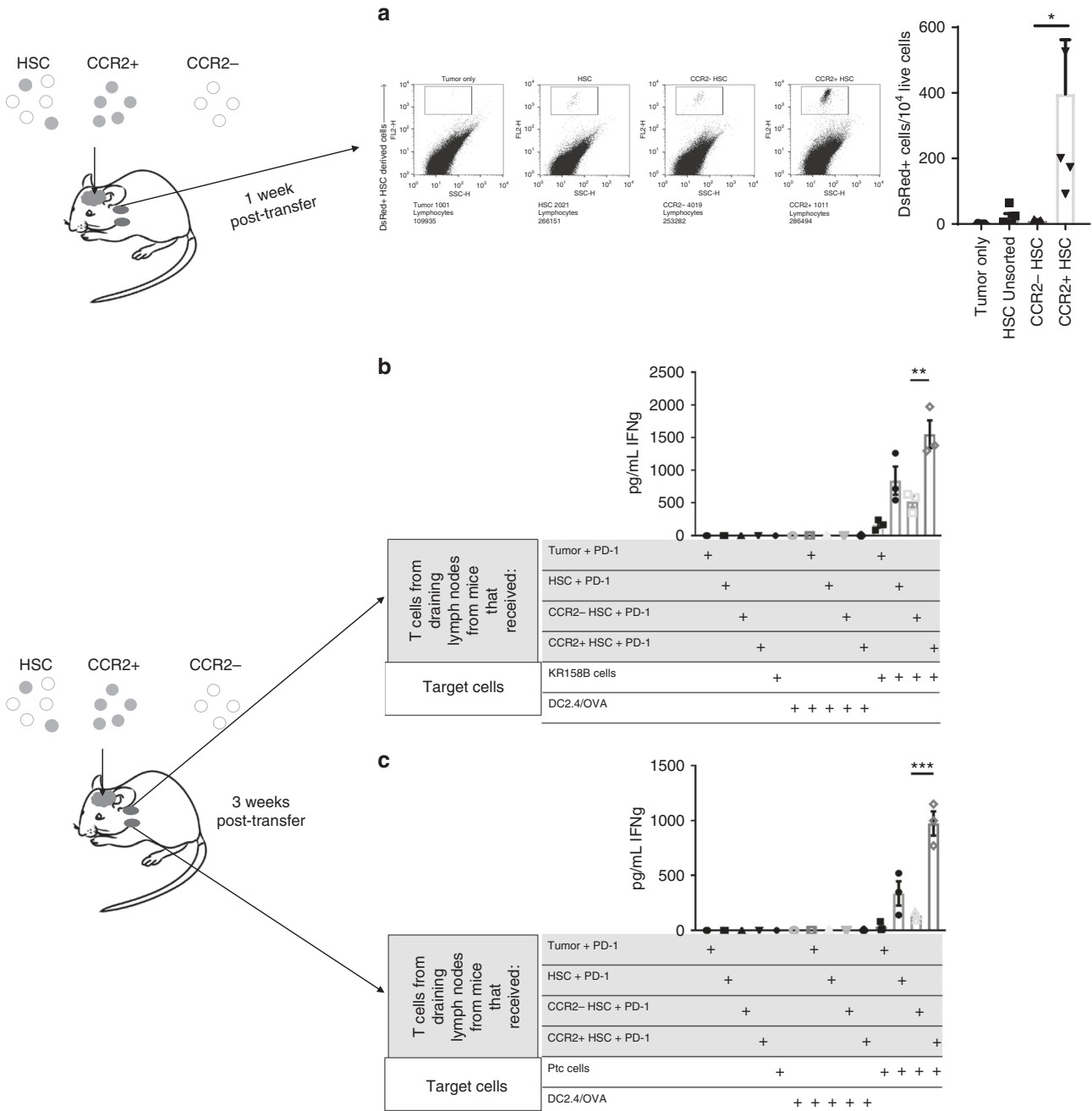

**Fig. 5** CCR2+HSCs cross-prime endogenous T lymphocytes. **a** Either HSCs, CCR2+HSCs, or CCR2−HSCs isolated from DsRed+ mice were stereotactically injected into tumors of glioma-bearing mice. DsRed+ cells were then observed in draining lymph nodes 1 week post transfer, and relative amounts of DsRed+ cells between groups was compared, CCR2+HSCs vs. CCR2−HSCs (*p = 0.021, Mann–Whitney, n = 5 mice per group). **b** Either HSCs, CCR2+HSCs, or CCR2−HSCs were isolated from DsRed+ mice, then were stereotactically injected directly into tumors of glioma- or medulloblastoma-bearing mice followed by systemic administration of PD-1. Three weeks post transfer, draining lymph nodes were harvested and T-cells were isolated from the lymph nodes using magnetic bead isolation. These T-cells were then used as effector cells against tumor cell targets in a co-culture assay. This was conducted in KR158B glioma-bearing mice, and Ptc medulloblastoma-bearing mice, and IFNγ secretion was measured after the co-culture. T cells harvested from **b** KR158B glioma-bearing mice that received CCR2+HSCs secreted an average of 1549.99 ± 212.621 pg/ml IFNγ, which is significantly more that T-cells derived from mice that received CCR2−HSCs that contained 522.183 ± 87.32 pg/ml IFNγ (**p = 0.0111, two-tailed t-test). **c** T cells harvested from Ptc medulloblastoma-bearing mice that received CCR2+HSCs secreted an average of 972.2 ± 110.6 pg/ml IFNγ, which is significantly more that T-cells derived from mice that received CCR2−HSCs that contained 132.6 ± 20.74 pg/ml IFNγ (***p = 0.0017, two-tailed t-test). All error bars represent s.e.m.

glioma-bearing mice received adoptive cell therapy with tumor-reactive T-cells and co-transfer of HSCs isolated from syngeneic GFP transgenic mice. Three weeks post adoptive cellular therapy, GFP+HSC-derived cells that migrated into brain tumor were isolated using FACS and used as target cells for in vitro expanded

effector T-cells. Supernatant IFNγ was measured and revealed that HSC-derived cells that were isolated from the tumor demonstrated specific presentation of tumor-derived antigens to activated T cells (Fig. 6a). To determine if HSC-derived cells have the capacity to present antigen to CD4+ or CD8+ tumor-specific

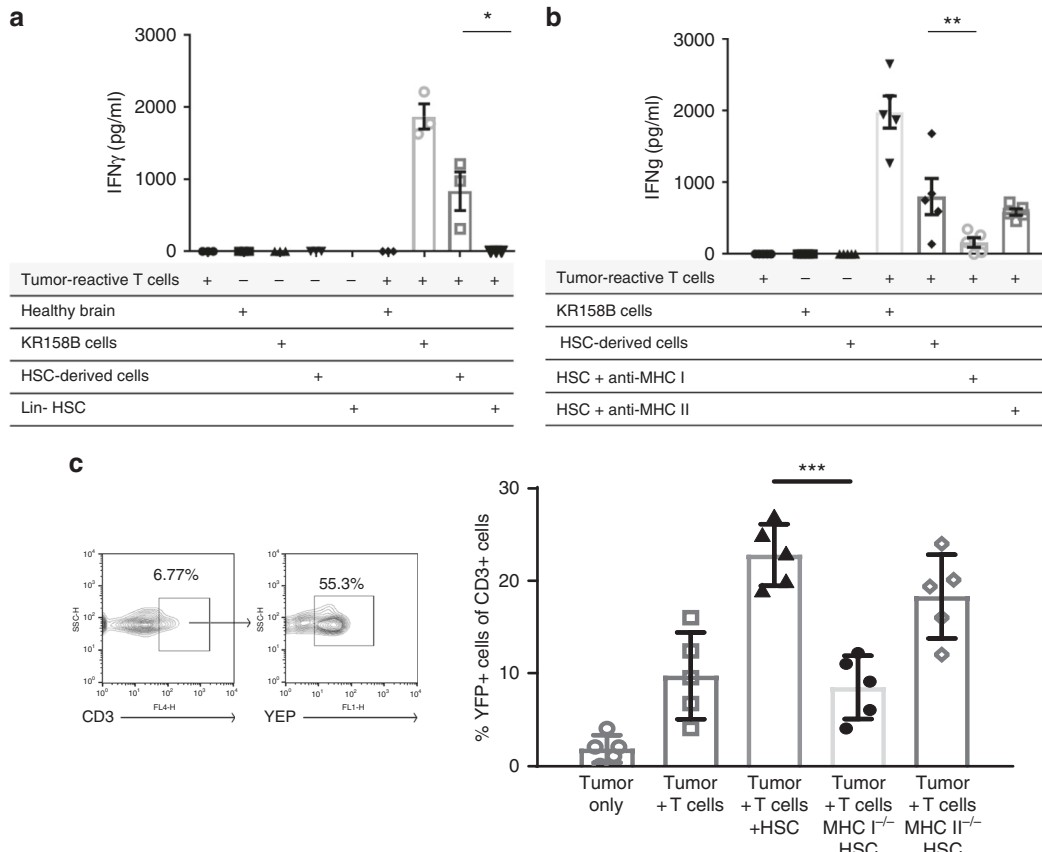

**Fig. 6** HSCs cross-prime tumor-reactive T-cells via MHC I. **a** Glioma-bearing mice received adoptive cell therapy with concomitant transfer of GFP⁺HSCs. Three weeks post transfer, GFP⁺ cells were FACS isolated from the tumor. These were then used as targets in a functionality co-culture assay for tumor-reactive T-cells, and supernatant IFNγ was measured. **b** In a repeat experiment, blocking antibodies for MHC I and MHC II were added to the co-culture assay and supernatant IFNγ was measured, and relative amounts between unblocked HSC-derived cell targets and when MHC-I was blocked (**$p = 0.0271$, t-test, $n = 3$ biological replicates). **c** Glioma-bearing mice received adoptive cell therapy with tumor-reactive T cells generated from GREAT mice followed by concomitant transfer of HSCs, HSCs from MHC I⁻/⁻, or HSCs from MHC II⁻/⁻. Tumors were excised and relative amounts of YFP⁺CD3⁺ cells in tumor was measured (***$p = 0.0002$, Mann–Whitney, $n = 5$ mice per group). All error bars represent s.e.m.

T cells, HSC-derived cells were again isolated with FACS and either MHC-I or MHC-II was blocked using blocking antibody for culture against effector tumor-specific T cells (Fig. 6b). After blocking MHC-I on HSC-derived cells, there was a significant decrease in IFNγ secretion observed relative to HSC-derived cells with no block ($p = 0.0271$). To confirm that HSC-derived cells present antigen via MHC-I to tumor-specific T cells, HSCs were isolated from MHC-I or MHC-II knockout mice and co-administered with adoptive cellular therapy. Here, the adoptively transferred tumor-reactive T cells were generated from GREAT mice which express YFP on the IFNγ promoter. Three weeks after adoptive transfer, tumors were harvested and tumor infiltrating lymphocytes were collected. T cells were analyzed for expression of YFP⁺CD3⁺ cells (Fig. 6c). A significant decrease in T-cell activation was detected in mice that received MHC-I⁻/⁻ HSCs vs. wildtype HSCs ($p = 0.0002$) but not in mice that received HSCs from MHC-II⁻/⁻ mice, demonstrating that cross-presentation of tumor antigen in the MHC-I pathway by HSC-derived cells is critical for in vivo T-cell activation within the CNS tumor microenvironment.

To further demonstrate the unique cross-priming capacity of APCs derived from CCR2⁺HSCs, we isolated antigen presenting cells directly from the tumors of mice receiving adoptive cellular therapy and used them as a cellular vaccine in recipient mice receiving DsRed⁺ tumor-reactive T cells. Tumor-bearing mice received adoptive cellular therapy and either GFP⁺HSCs,

GFP⁺CCR2⁺HSCs, or GFP⁺CCR2⁻HSCs (Fig. 7a). Three weeks post adoptive cellular therapy, GFP⁺ cells from all tumors were harvested and isolated using FACS. These were then used as a vaccine in a second cohort of tumor-bearing mice that received adoptive cellular therapy. This cohort received either no vaccine, dendritic cell vaccine, or APCs isolated from the tumor microenvironment derived from GFP⁺HSCs, GFP⁺CCR2⁺HSC-derived cells, or GFP⁺CCR2⁻HSC-derived cells. One week later, vaccine-site draining lymph nodes were harvested and analyzed for relative expansion of DsRed⁺CD3⁺ T cells and demonstrated that CCR2⁺HSCs led to the expansion of tumor-reactive T cells in vivo (Fig. 7b). Collectively, these experiments demonstrate that CCR2⁺HSCs uniquely give rise to APCs that capture tumor antigen in vivo and cross-present tumor antigens to CD8⁺ T cells in vitro and in vivo.

Earlier, we also demonstrated that CCR2⁺HSC-derived cells cross-present tumor antigen when co-transferred with PD-1 (Fig. 5b, c). Next, we conducted experiments to confirm that CCR2 + HSC-derived cells cross-present antigen via MHC-I during combinatorial therapy with PD-1. To do this, GREAT mice were implanted with intracranial gliomas then treated with PD-1. We then isolated CCR2⁺ and CCR2⁻HSCs from bone marrow of wildtype C57BL/6 mice, MHC-I⁻/⁻ mice, and MHC-II⁻/⁻ mice. GREAT mice bearing late-stage intracranial tumors were then treated with PD-1 and either no treatment, PD-1 only, CCR2⁺HSC + PD-1, CCR2⁺MHC-I⁻/⁻ HSC + PD-1, CCR2⁺MHC-II⁻/⁻ HSC + PD-1,

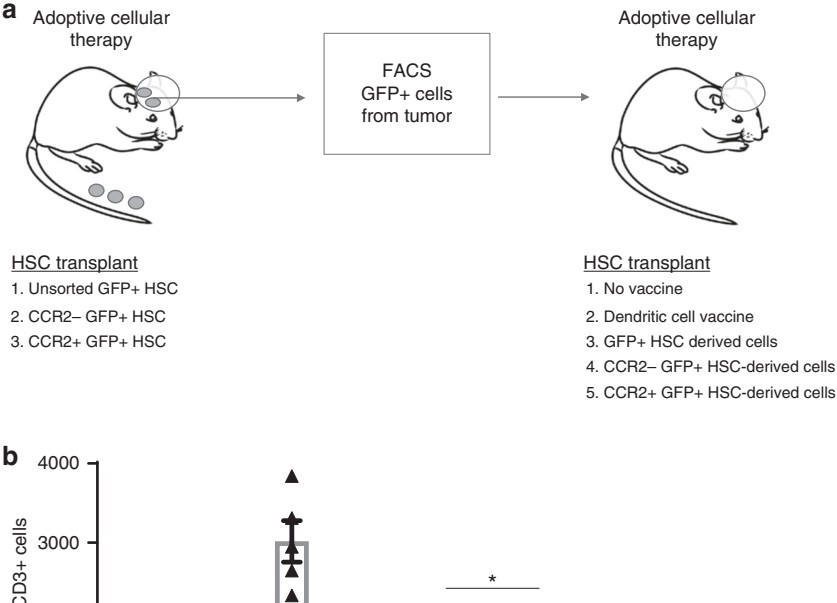

**Fig. 7** CCR2$^+$HSC-derived dendritic cells expand T-cells in adoptive cell therapy. **a** Experimental description. **b** Cells derived from GFP$^+$HSCs, GFP$^+$CCR2$^-$HSCs, or GFP$^+$CCR2$^+$HSCs were isolated from the tumors of mice treated with adoptive cell therapy. These were then used as a vaccine in a second cohort of mice treated with adoptive cell therapy using DsRed + tumor-reactive T cells. Vaccine draining lymph nodes were harvested and relative amounts of DsRed$^+$ T-cells were measured between the group that received CCR2$^+$HSCs and the groups that received cells derived from CCR2$^-$HSCs (*$p = 0.0001$, $t$-test, $n = 3$ replicates per group). All error bars represent s.e.m.

CCR2$^-$HSC + PD-1, CCR2$^-$MHC-I$^{-/-}$ HSC + PD-1, or CCR2$^-$MHC-II$^{-/-}$ HSC + PD-1 (Fig. 8a). One week after treatment, tumors were excised and processed into single-cell suspensions. Flow cytometric analysis was conducted to determine expression of YFP on CD3$^+$ cells. Results demonstrate significantly more YFP$^+$CD3$^+$ cells in mice that received CCR2$^+$HSC + PD-1 (mean YFP$^+$CD3$^+$ cells = 14.41% ± 1.729, $n = 5$) than PD-1 alone (mean YFP$^+$CD3$^+$ cells = 3.58 ± 0.945, $n = 5$; $p = 0.0006$) (Fig. 8a). The relative amount of YFP$^+$CD3$^+$ cells in tumor was also significantly decreased in the group that received CCR2$^+$MHC-I$^{-/-}$ HSC + PD-1 (mean YFP$^+$CD3$^+$ cells = 0.597% ± 0.2487, $n = 5$) relative to the group that received CCR2$^+$HSC + PD-1 ($p = <0.0001$). Interestingly, we also observe a significant decrease in YFP$^+$CD3$^+$ cells in the group that received CCR2$^+$MHC-II$^{-/-}$ HSC + PD-1 (1.196% ± 0.5061, $n = 5$, $p = 0.0001$), suggesting at the least, that presentation to both CD4 and CD8 T cells are necessary for the observed T-cell activation. We also observed no significant increase in YFP$^+$CD3$^+$ within the tumor in any groups that received CCR2$^-$HSCs relative to the no treatment group. This further confirms our observations that CCR2$^+$HSCs are responsible for increased activation of tumor-infiltrating T-cells.

**Transfer of CCR2$^+$HSCs increases efficacy of immunotherapy.** The therapeutic effect we observed above (Fig. 1a, b) occurred with combinatorial HSC + PD-1 therapy. To determine if CCR2$^+$HSCs are the subset of cells responsible for the observed efficacy of the combinatorial therapy, we compared the therapeutic

effect of bulk HSC + PD-1 vs. CCR2$^+$HSC + PD-1 in tumor-bearing mice. We evaluated this against both the KR158B glioma and the Ptc medulloblastoma (Fig. 8b, c). In glioma-bearing mice, the combination of purified CCR2$^+$HSCs + PD-1 led to significantly increased median survival and long-term survivors over the PD-1 alone group ($p = 0.0006$), as well as the cohort that received bulk HSCs + PD-1 ($p = 0.0233$)(Fig. 8b). In mice bearing Ptc medulloblastoma, combinatorial CCR2$^+$HSCs + PD-1 also provided a significant survival benefit over both PD-1 only ($p = 0.0001$) and bulk HSC + PD-1 ($p = 0.0005$) (Fig. 8c).

To determine if lineage negative CCR2$^+$HSCs are true hematopoietic stem cells that provide rescue from bone marrow failure, CCR2$^+$HSCs or CCR2$^-$HSCs were intravenously administered into myeloablated hosts (9 Gy total body irradiation). CCR2$^+$HSCs were not as efficient in rescuing hosts from lethal irradiation, indicating enrichment of a progenitor population rather than a multipotent stem cell population (Fig. 8d). To determine if CCR2$^+$HSCs are responsible for the enhanced efficacy of stem cell transfer in adoptive cellular therapy targeting brain tumors, purified CCR2$^+$HSCs vs. CCR2$^-$HSCs were transferred in conjunction with adoptive cellular therapy in mice receiving non-myeloablative conditioning (5 Gy total body irradiation). CCR2$^+$HSCs were markedly superior in enhancing the efficacy of adoptive cellular therapy against glioma relative to bulk unsorted HSCs ($p = 0.0005$) (Fig. 8e). This was repeated in our model of medulloblastoma which demonstrated that CCR2$^+$HSCs are responsible for the capacity of HSC transfer to enhance the efficacy of adoptive cellular therapy (Fig. 8f). These

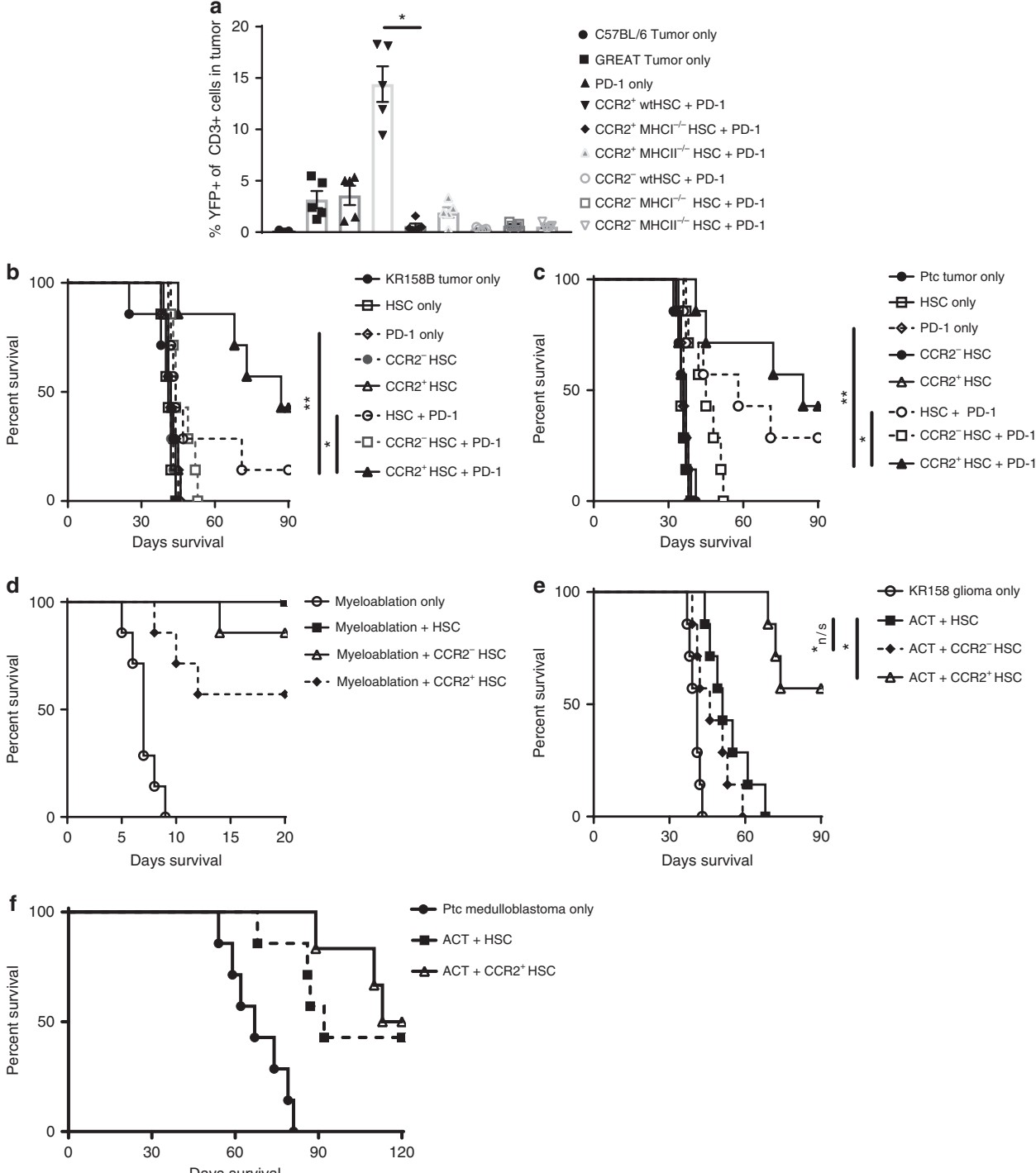

**Fig. 8** CCR2+HSCs increase efficacy of PD-1 against brain tumors. **a** KR158B tumor-bearing GREAT mice were treated with PD-1 and either CCR2−HSCs or CCR2+HSCs isolated from either wildtype C57BL/6 mice, MHC I−/− mice, or MHC II−/− mice. Tumors were excised and relative amounts of YFPP+CD3+ cells were measured, specifically between the group that received wildtype CCR2+HSCs and CCR2+HSCs from MHC I−/− (*$p < 0.0001$, Mann–Whitney, $n = 5$ mice per group). **b** KR258B tumor-bearing mice received no treatment, HSC only, PD-1 only, CCR2−HSCs, CCR2+HSCs, HSCs+PD-1, CCR2−HSCs + PD-1, or CCR2+HSCs + PD-1 (*$p = 0.0233$, **$p = 0.0006$). **c** Ptc medulloblastoma tumor-bearing mice received no treatment, HSC only, PD-1 only, CCR2−HSCs, CCR2+HSCs, HSCs + PD-1, CCR2−HSCs + PD-1, or CCR2+HSCs + PD-1 (*$p = 0.0001$, **$p = 0.0005$). **d** Mice that received lethal total body irradiation for myeloablative host conditioning received either no treatment, HSCs, CCR2−HSCs, or CCR2+HSCs, then followed for survival. **e** KR158 glioma-bearing mice received adoptive cell therapy (ACT) and either HSCs, CCR2−HSCs, or CCR2+HSCs, *$p = 0.0005$, *n/s $p = 0.1776$. **f** Ptc medulloblastoma-bearing mice received adoptive cellular therapy (ACT) and either HSCs or CCR2+HSCs. All error bars represent s.e.m. Comparison of survival data was conducted with Gehan–Breslow–Wilcoxon test

findings are the first to identify a bone marrow-derived progenitor cell population that has the capacity to alter the tumor microenvironment and enhance anti-tumor responses to both immune checkpoint blockade and adoptive cellular therapy. These findings alter our understanding of the role of stem cell transplantation in the treatment of solid malignancies and hold relevance for addressing resistance to cancer immunotherapy.

## Discussion

The most common application of HSC transplantation is to regenerate the hematopoietic system and to restore deficient hematopoietic functions. In routine clinical settings, HSC transfer is typically used for rescue of bone marrow after high-dose chemotherapy or radiation exposure[16]. Clinical studies have explored the capacity of HSC transfer as treatment for life-threatening immunodeficiencies, congenital disorders, as well as for the induction of tolerance prior to organ transplant. In all these cases, HSCs are leveraged for their capacity to correct abnormalities and promote homeostasis between the host and transferred cells. In contrast, there are only two situations that we are aware of leveraging HSC transfer to purposefully enhance immunologic responses, both of which are in oncology[16]. The first is in the context of allogeneic HSC transplant, where graft-vs.-tumor response was found to control relapse of hematological malignancy[16]. In these immunologic events, donor-derived immune cells attack recipient tissue, including cancer cells, thereby delaying relapse. In this context, however, the graft-vs.-tumor effects are believed to be mediated predominately by donor lymphocytes contained within the allograft recognizing minor histocompatibility differences and/or lymphocytes that differentiate from transplanted stem cells and are primed in the host against tumor-derived antigens. The stem cells have not been demonstrated to play a primary role in tumor rejection in this context.

In this study, we unraveled a novel role for HSCs in potentiating endogenous anti-tumor responses when co-transferred with either PD-1 inhibitor or adoptive cell therapy in syngeneic murine models of brain tumors. A proportion of co-transferred HSCs migrate to intracranial tumor, differentiate into dendritic cells, and subsequently lead to the maintenance of anti-tumor T-cell activation within the tumor microenvironment.

Concomitant HSC transfer with PD-1 not only led to increased intra-tumor T-cell activation, but as shown in Fig. 1e, down-regulated gene expression of multiple immunosuppressive pathways within the tumor microenvironment after a single infusion of cells. A phase III trial evaluated the efficacy of anti-PD-1 monoclonal antibody in recurrent glioblastoma and demonstrated that these tumors are resistant against PD-1 immunotherapy[3]. Here we leverage the use of HSC transfer to reverse the resistance to PD-1 checkpoint blockade in murine brain tumor models refractory to PD-1 treatment alone. Our previous study demonstrated that HSC transfer enhances the efficacy of adoptive T-cell therapy against gliomas[5,6]. In both studies, the impact of HSCs on the function of tumor infiltrating lymphocytes was paramount in the enhancement of anti-tumor activity[6]. Brain tumors are notoriously immunosuppressive, shutting down T-cell activation. We have found that adoptively transferred HSCs traffic to intracranial tumors where they not only increase accumulation of activated T cells[5], but also differentiate into dendritic cells and subsequently cross-prime both endogenous and adoptively transferred tumor-reactive T cells. We believe that the maintenance of T-cell activation within the tumor microenvironment plays a major role in the observed efficacy against otherwise refractory tumors.

The HSCs used in these studies are a bone marrow-derived lineage negative, heterogeneous population of stem and progenitor cells. Herein, we identify the progenitor cells responsible for our observations as CCR2$^+$Lin$^-$ cells (CCR2$^+$HSCs). CCR2 is expressed by monocyte progenitor cells and is required for their entrance into the CNS[12,13]. We demonstrate in Fig. 8d that CCR2$^+$HSCs have reduced capacity to rescue bone marrow of mice after myeloablative total body irradiation. CCR2 is also expressed by MDSCs which are strongly associated with poor outcomes in glioblastoma. Importantly, we demonstrate that the cells derived from the transferred CCR2$^+$HSCs actually downregulate the expression of CCR2 within the tumor. We believe that CCR2 expression by CCR2$^+$HSCs is maintained until the cells cross the blood-brain barrier, but is then either downregulated or shed within the tumor microenvironment once they have differentiated into antigen presenting cells. We recently demonstrate that HSC differentiation into antigen presenting cells is driven by IFNγ from activated T cell within the tumor microenvironment[6]. The APCs derived from CCR2$^+$HSCs do not stay in the brain tumor, but extravasate into secondary lymphoid organs and cross-present tumor antigens to endogenous T cells. This is the first study to demonstrate that antigen can be captured within intracranial brain tumor and cross-presented to T cells in secondary lymphoid organs by HSC-derived APCs.

In contrast to the immune potentiating effects of CCR2$^+$HSCs, it is very interesting that the CCR2$^-$HSCs that reach the intracranial brain tumor actually upregulate CCR2 expression along with markers of MDSCs including CD11b, Ly6G/C, and F4/80 (Fig. 4c, d and Supplementary Fig. 1). These cells are incapable of enhancing responses to PD-1 blockade or adoptive cellular therapy, and are consistent with progeny with immunosuppressive functions. The differential roles of CCR2$^+$ and CCR2$^-$ HSCs is an area of ongoing study at our center.

The advance of checkpoint inhibitors in the treatment of advanced solid tumors is quite remarkable, and yet many patients remain refractory to monotherapy. The capacity to increase clinical responses has been demonstrated through combinatorial approaches that combine checkpoint inhibitors with other immunotherapeutic modalities or targeted therapeutics. Currently hundreds of studies are exploring combining PD-1 blockade with multiple pharmacologic agents. These ongoing efforts have the challenge of dealing with combinatorial toxicities as well redundant immunoregulatory pathways that may overcome the blockade of one or more specific pathways. One of the particularly attractive features of the HSC-mediated sensitization strategy we have revealed is the observation that multiple immunoregulatory pathways are modulated with HSC infusion (Fig. 1e). Given the well-established tolerance of HSC transfer in humans, this approach may afford a combinatorial strategy that can impact on multiple immune checkpoint pathways simultaneously, with potentially minimal additional toxicity. The clinical advancement of immunomodulatory stem cell therapy is an area of active exploration and focus of future research.

## Methods

**Mice.** Female 6- to 8-week-old C57BL/6 mice (Jackson Laboratories stock 000664), transgenic DsRed mice (Jackson Laboratories stock 006051), transgenic GFP mice (Jackson Laboratories stock 004353and transgenic GREAT mice (Jackson Laboratories stock 017580) were used for experiments. The investigators adhered to the "Guide for the Care and Use of Laboratory Animals" as proposed by the committee on care of Laboratory Animal Resources Commission on Life Sciences, National Research Council. The facilities at the University of Florida Animal Care Services are fully accredited by the American Association for Accreditation of Laboratory Animal Care, and all studies were approved by the University of Florida Institutional Animal Care and Use Committee.

**RNA isolation.** Total tumor RNA isolation from tumor cell lines was performed with RNeasy mini kit (Qiagen, cat. 74104) as per the manufacturer's protocol.

**Hematopoetic stem cells**. Fresh bone marrow is harvested from female 6- to 8-week-old mice. Red blood cells are lysed, and mononuclear cells are counted and collected. Lineage-negative cells are isolated using magnetic bead isolation kit (Miltenyi Biotec, cat. 130-090-858) as per manufacturer's protocol. MNCs are incubated with Biotin−Antibody Cocktail (10 μl per $10^7$ total cells) for 10 min, followed by incubation with Anti-Biotin MicroBeads (20 μl per $10^7$ total cells). After PBS wash, the mixture is then resuspended in buffer and placed through a magnetic separation column for separate lineage-negative cells. To further isolate the Lin- HSCs to stem cell subsets, the lineage negative population was then labeled with either anti-Sca-1-biotin (10 μl per $10^7$ MNC, Miltenyi cat. 130-101-885), CD117 MicroBeads (20 μl per $10^7$ MNC, Miltenyi cat. 130-091-224), anti-Prominin-1-biotin (5 μl per $10^6$ MNC, Miltenyi cat. 130-101-851), anti-CD38-biotin (5 μl per $10^6$ MNC, Miltenyi cat. 130-109-253), or anti-CCR2−biotin (5 μl per $10^6$ MNC, Miltenyi cat. 130-108-721), incubated for 10 min at 4 °C, then 20 μl of anti-biotin microbead was added per $10^7$ MNC, and incubated for 15 min at 4 °C. Mixture was then washed with PBS and centrifuged at $500 \times g$ for 5 min. Cells were resuspended in 500 μl buffer per $10^8$ MNC and placed through magnetic column.

**Tumor models**. Tumor-bearing experiments were performed in syngeneic sex-matched C57BL/6 mice. KR158B[11] gliomas were supplied by Dr. Karlyne M. Reilly at the National Cancer Institute, Bethesda, MD. Ptc medulloblastoma line was provided in collaboration with Dr. Robert Wechsler-Reya at the Sanford Burnham research Institute, La Jolla, CA. This line is maintained in vivo and checked annually for genetic markers consistent with sonic hedgehog molecular subtype medulloblastoma[4]. For experiments using glioma, $10^4$ KR158B murine high-grade astrocytomas were implanted into the caudate nucleus by injecting 2 mm lateral to the midline and 3 mm deep[5,11]. For experiments using Ptc medulloblastoma, $1.25 \times 10^5$ Ptc cells were implanted into the cerebellum 1 mm lateral to the midline and 3 mm deep[4,17]. Tumors were injected with a stereotactic frame (Stoelting, cat. 53311) and a 250 μl syringe (Hamilton, cat. 81120) with a 25-gauge needle for KR158B-luc tumors and a 5 μl syringe (Hamilton, cat. 88011) for Ptc tumors. All tumor lines are tested annually for pathogens by IDEXX BioResearch (Westbroon, ME).

**Tumor-reactive T cells for adoptive cellular therapy**. Bone marrow was harvested from C57BL/6 mice, red blood cells were lysed and MNCs were cultured in GM-CSF (10 ng/ml, R&D, cat. 415-ML/CF) and IL-4 (10 ng/ml, R&D, cat. 404-ML/CF) for 9 days. Dendritic cells were then electroporated with 25 μg total RNA isolated from tumor tissue. Naïve mice were primed with $2.5 \times 10^5$ total tumor RNA-pulsed dendritic cells. After one week, splenocytes were then harvested and co-cultured ex vivo using total tumor RNA-pulsed dendritic cells and IL-2 (50 U/ml, R&D, cat. 402−ML/CF) for 5 days. $10^7$ TTRNA-T cells were intravenously administered for adoptive cell therapy.

**Adoptive cellular therapy**. Treatment of tumor-bearing mice began with 5 Gy lymphodepletion or 9 Gy myeloablation on day 4 post intracranial injection with X-ray irradiation (X-RAD 320, PXINC). On day 5 post intracranial tumor injection, mice received a single intravenous injection with $10^7$ autologous ex vivo expanded TTRNA T cells with either $5 \times 10^4$ lineage-depleted (lin−) hematopoietic stem and progenitor cells (HSCs) (Miltenyi Biotec cat. 130-090-858), CCR2+HSCs, or CCR2−HSCs. CCR2 positive selection was conducted using biotinylated anti-mouse CCR2 antibody (Miltenyi Biotec cat. 130-108-721) followed by anti-biotin Microbead separation (Miltenyi Biotec clone Bio3-18E7.2). Beginning day 7 post-tumor injection, $2.5 \times 10^5$ tumor RNA-pulsed dendritic cell vaccines were injected intradermally posterior to the ear pinna weekly for three total vaccine doses.

**Mouse lymph node digestion**. A minimum of two lymph nodes were dissected bilaterally from the cervical region of treated mice. Lymph nodes were mechanically dissociated with a sterilized razor blade (Personna cat. 94-120-71) and chemically digested with 2% collagenase (Fisher Scientific cat. 10103578001) for 30 min.

**Brain tumor digestion**. Brain tumor dissection began posteriorly with a midline cut in the skull and rongeur removal of skull laterally. Tumor resection extended to gross borders of tumor mass near the site of injection. Tumors were dissociated mechanically with a sterilized razor blade and chemically with papain (Worthington cat. NC9809987) for 30 min. Tumors were filtered with a 70 μm cell strainer (BD Biosciences cat. 08-771-2) prior to antibody incubation.

**Flow cytometry**. Flow cytometry was performed on FSC/SSC gating on the BD Biosciences FACSCanto II. IFN-γ release by cells from transgenic GREAT mice were detected at FL-1 and cells from transgenic DsRed mice were detected at FL-2 and FACS sorted using the BD Biosciences FACSAria II. Cells were prepared ex vivo as described above and suspended in 2% FBS (Seradigm cat. 97068-091) in PBS (Gibco cat. 10010−049). Antibodies below were applied per manufacturer's recommendation with isotype controls: APC-conjugated anti-CD3 (1:50 dilution, BD cat. 553066), APC-conjugated anti-CD11c (1:100 dilution, Affymetrix cat. 17-0114-82), anti-CD80 (1:50 dilution, Affymetrix cat. 17-0801-82), anti-CD86 (1:50

dilution, Affymetrix cat. 17-0862−82), anti-Ly-6G/6 C (1:100 dilution, BD Biosciences cat. 553129), anti-F4/80 (1:50 dilution, eBioscience cat. 17-4801-80), and anti-MHC II IA-E (1:100 dilution, Affymetrix cat. 17-5321-82). Analysis and flow plots were generated with FlowJo version 10 (Tree Star).

**T-cell function assay**. In vitro experiments utilized IFN-γ release as a measure of T-cell activity after a restimulation assay in which effector cells and targets are co-cultured in a 10:1 ratio in 96-well U-bottom plates in triplicate. IFN-γ Platinum ELISAs (Affymetrix, catalog BMS606) were performed on harvested and frozen acellular media from the supernatants of the 96-well co-culture plates after 1 day of co-culture. Anti-mouse MHC Class II (I-A/I-E) blocking antibody was purchased from Affymetrix (cat. 16-5321-85). Anti-mouse MHC Class I (H-2K) blocking antibody was purchased from Affymetrix (cat. 16-5957-85).

**Anti-PD-1 blocking antibody**. Administration of anti-PD-1 blocking antibody (Merck mDX-400, or Bxcell RMP1-14) began on the day of T-cell administration and continued every 5 days for a total of four doses of 10 mg/kg.

**PCR array**. PCR analysis was performed on tumors excised from treated mice. Tumors were dissociated and RNA isolated as described above and analyzed with the $RT^2$ Profiler Array Cancer Inflammation and Immunity Crosstalk (Qiagen cat. PAMM-181ZD-12) or T-cell and B-cell activation (Qiagen cat. PAMM-053ZD-2) as per manufacturer's protocol.

**Statistical analysis**. All experiments were analyzed in Prism 7 (GraphPad) and tests were applied as described in the figure legends. The log-rank test was utilized to compare Kaplan-Meier survival curves. An unpaired, Mann–Whitney rank-sum test was applied for two-group comparisons for in vivo experiments. An unpaired, Student's $t$-test was applied for two-group comparisons for in vitro experiments. The data have normal distribution and variance was similar between groups statistically compared. Significance is determined as $p < 0.05$. For animal studies where tissue was analyzed for biological endpoints, $n = 5$ mice per group and no statistical methods were used to determine sample size. The authors pre-established that no animals or samples were to be excluded from analysis. For randomization of animal experiments, mice were housed at 5 mice per cage. After tumor implantation, mice were immediately randomized to cages. Mice were again randomized to cages after total body irradiation. No other blinding was conducted in these studies.

## Data availability
All relevant data are available from authors.

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

## Acknowledgements

Data reported are available in the paper or in Supplementary Information. This research was supported by the Florida Center for Brain Tumor Research/Accelerate Brain Cancer Cure Grant (C.T.F.), Alex's Lemonade Stand Foundation Young Investigator Award (C.T.F.), American Brain Tumor Association Collaboration Grant (C.T.F.), Michael Mosier Foundation Defeat DIPG Chad Tough Research Grant (C.T.F.), NIH/NCI R01–CA195563 (D.A.M.), and the Preston A. Wells, Jr. Center for Brain Tumor Therapy at the University of Florida.

## Author contributions

C.T.F. performed the experiments and analysis with assistance from T.J.W., R.S.A., G.L.M., B.D.D., J.A.D. D.A.M. supported the project. All authors wrote or edited the manuscript.

## Additional information

**Competing interests:** C.T.F. and D.A.M. hold patents related to the content of this manuscript and are co-founders and equity holders in iOncologi, Inc., and as such may benefit financially as a result of the outcomes of their research or work reported in this publication. The remaining authors declare no competing interests.

