## [Peer Review File · Nature Communications]

Reviewers' comments:

Reviewer #1 (Remarks to the Author):

This manuscript focuses on the ability of hematopoietic stem cells (HSCs) to enhance immune responses to brain tumors. Using syngeneic mouse models of glioblastoma (GBM) and medulloblastoma (MB), the authors demonstrate that HSCs expressing CCR2 can enhance the effects of PD-1 blockade or adoptive T cell therapy. They show that intravenously administered CCR2+HSCs can migrate to intracranial tumors and differentiate into CD11c+ antigen presenting cells (APCs) at the tumor site. These APCs can present tumor-derived antigens to CD4+ and CD8+ T cells, leading to T cell activation and enhanced tumor rejection. Based on these studies, they conclude that CCR2+HSCs could be used to overcome resistance to PD-1 blockade and adoptive cellular therapy.

Overall, this is an extremely interesting paper that has important implications for therapy of brain tumors. The experiments are generally well designed and controlled, and the conclusions are supported by the data. My main concern is that the paper is very difficult to follow. There are many places where figures are not properly labeled, terms and abbreviations are not well defined, and experimental methods are not described in sufficient detail. In addition, a large proportion of the important data is relegated to supplemental figures, which is not a problem for reviewers but will make the paper difficult to read if and when it is published. These issues are prevalent enough that they interfere with interpretation of the results and appreciation of their significance. While very few new experiments are necessary, a cleaned-up, simplified, user-friendly version of the manuscript should be provided. Detailed comments and questions are listed below.

1. The Abstract and Introduction seem redundant. Since the Abstract summarizes the key findings of the paper, it would be helpful if the Introduction was used to present relevant background that is necessary to understand the significance of these studies
2. The manuscript contains virtually no Discussion section. Notwithstanding the word limits, it would be valuable to include at least a paragraph or two discussing the implications of this work for the field. One question that should be addressed – either experimentally or through discussion – is why endogenous HSC's don't migrate to the CNS tumor microenvironment and mediate the same effects as the transplanted HSCs in these studies
3. A large proportion of the data is presented in Supplementary Figures. Since much of this information is critical to the logic of the paper, I would suggest incorporating some of the data back into the main figures. The supplements should be reserved for data that are less central to the argument of the paper, but provide key support for the main experiments.
4. Like the main figure legends, the supplemental figure legends should have titles that highlight the point of the figure, rather than just launching into the details of the experiment.
5. On p. 4, the authors refer to patched tumors as representing Group 3 medulloblastoma. This is incorrect; they represent sonic hedgehog-driven medulloblastoma.
6. In many of the figures, the authors refer to experimental groups as "tumour only", "HSC only," "Anti-PD-1 only," etc. This is confusing: they should either refer to the tumour-only group as "no treatment", or refer to the treatment groups as "tumour + HSCs," "tumour + Anti-PD-1", etc
7. Many of the FACS plots (e.g. Supplemental Figures 4, 5 and 8) are very difficult to read. The Y-axes are illegible, and there are no numbers shown to indicate the percentages of cells in the red boxes. Without this information, the reader cannot assess the results properly.
8. The authors should indicate how many replicates they did for each and every experiment in the paper. Statements like "this has been repeated in subsequent experiments" are not sufficient
9. The authors do not provide enough detail regarding their method for purifying HSCs; they only mention this in the methods, referring to a kit from Miltenyi. Since HSCs are the focus of the entire manuscript, it would be helpful to explain this in detail, and to show FACS plots verifying the purity of this population.
10. In Figure 1A, it is noteworthy that about half the mice treated with HSC + Anti-PD-1 die with kinetics similar to the other treatment groups, and the other half show prolonged survival. This

bimodal effect is interesting, and suggests that only some mice are responsive to HSCs. If this is a reproducible observation, the authors should comment on it.

11. In Figure 2A, the effect of unsorted HSCs + anti-PD-1 is much weaker than in Figure 1A. Which is a more representative result?

12. In Supplemental Figure 3, it is not clear what the numbers across the top and the letters on the left side represent. Do different rows (or columns) represent different mice? Or is the whole heat map a representation of the ratio of expression in one group of mice versus the other? This needs to be clarified in the legend. And rather than simply pointing to a few of the spots on the array, the identity of all the spots should be shown in a table or list accompanying the figure.

13. For the experiment in Supp Figure 4, based on the x-axis label, the authors seem to use dsRed to identify donor-derived cells. This is a critical element of the experiment, but it is not mentioned in the text or the legend. The details of this experiment should be clarified.

14. For Supp Figure 5, was the phenotype of the dsRed+ cells in the lymph nodes determined? Do the authors know if these cells included dendritic cells, macrophages, or even T cells? This information is important for interpreting the results of this experiment.

15. The PCR array in Supp. Figure 6 has the same issues as the one in Supp. Figure 3. In addition, the significance of the arrow pointing to "IFN?" is unclear. Are they not sure that this spot represents interferon? Or is the "?" supposed to be a gamma? This needs to be corrected. And all the probes on the array should be identified, rather than just pointing out the one that they are interested in.

16. On p. 10, discussing Supp. Figure 7C, the authors say "To determine if these HSC subsets migrate to intracranial tumor, DsRed+CCR2+HSCs and GFP+CCR2-HSCs were intravenously injected in equal amounts (5x10⁵ cells) into lymphodepleted tumor-bearing mice." How is this experiment different from the previous experiments involving dsRed-expressing HSCs (e.g. Supplemental Figure 4)? Haven't they already established that the cells can migrate to the brain? If the experiments are redundant, one should be removed. If they are complementary, this should be clarified, and the authors should consider combining these data into one figure.

17. In the text (pp. 10-11) describing Supp Fig 8, the authors state that "Antigen presenting cells from CCR2+HSCs had a distinct dendritic cell phenotype while cells that arose from CCR2-HSCs upregulated expression of monocyte suppressor cell marker Ly6G (Gr1-1)". Is the dendritic cell phenotype determined only by IA-E? The legend to Supp Figure 8 mentions a number of other markers (CD11c, CD80, CD86), but the data for these markers are not presented. The authors need to provide stronger evidence for the DC vs. monocyte phenotype

18. In Supp Figure 9A there is no explanation for "Brain" and "Kluc cells" and no axis or label that explains the cells listed above the line (are these effectors?) and below the line (targets)?

19. It is not clear what the FACS plot in Supp Figure 9C is showing

20. The experiment depicted in Supp Fig 10B needs to be explained more clearly.

21. In Figure 3, there is no explanation for the term "MA"

22. The brain stem glioma cells used in Supp Figure 11 are not described in any detail. What mouse model are they from?

Reviewer #2 (Remarks to the Author):

The study titled "Lin-CCR2+ hematopoietic stem cells overcome resistance to PD-1 blockade in brain tumors" demonstrates that bone marrow-derived, lineage-negative hematopoietic stem and progenitor cells (HSCs) that express CCR2+ reverse treatment resistance and sensitizes mice to curative immunotherapy. Although authors have performed extensive survival based studies in mice bearing different intracranial tumor types with appropriate controls, fundamental questions with regards to the fate of HSC, their delivery via different routes of administration and the rationale of combining the Lin- CCR2+ HSC with anti-PD1 therapy is not presented. Specifically, the authors claim enhanced recruitment of Lin-CCR2+ HSCs, however, detailed analysis of the HSC recruitment within the whole tumors has not been studied. Labelling HSC with luciferase should help following the recruitment of systemically delivered HSC. Additionally, whether or not these

cells have better differentiation capabilities than the CCR2-Lin- HSCs needs to be more clearly established.

The authors do not provide a rationale for combining it with PD-1 therapy. As observed by the authors, Lin-CCR2+ HSCs are not very efficient in rescuing hosts from lethal irradiation, these cells appear to be a 'more- developed' (and potentially developed) from their lin-CCR2- HSC counterparts (Fig. 3A). This is also supported by the finding that Lin-CCR2- HSCs become Lin-CCR2+ HSCs post-transfer (suppl Fig 4). It can be hypothesized that author's observations have a temporal effect: CCR2-Lin-HSCs become CCR2+ and mature to have better antigen presenting efficacy. Time kinetics experiments are needed to fully elucidate that CCR2+ HSCs are better than CCR2- HSCs. The finding that these CCR2+ HSCs-derived cells can home better in lymph nodes is not surprising as inflamed lymph nodes (as in the case of intracranial tumor implantation) have higher CCL2 secretion and 2 distinct blood monocytic populations (CCR2+ and CCR2-) have been well characterized.

Other concerns:

1.The abstract states "but these findings are restricted to occur only within intracranial brain tumors." What is the rationale behind the claim of brain tumor specific therapeutic efficacy? I don't see any other tumor models tested in this study.

2.Supplement Fig 4:

a.pre-transfer characterization of the population of HSCs before being transferred is essential (wt LY6C, MHC-II and CD11c) to demonstrate their antigen presenting potential.

b.Do not see DS-Red negative Ly6G/C+ve cells?

c.Provide control where no HSCs were administered.

d.Please comment on CCR2-ve cells have become CCR2+ve and vice-versa

3.Figure 1 shows 40-50% survival in anti-PD-1+HSC group while in Figure 2, similar survival is reported for PD-1 treatment+CCR2+HSC. For PD-1+HSC group, Survival is only 20% in Figure 2 experiment? There seems to be variation in data obtained from different experiments?

4.If the authors claim that CCR2+Lin-HSCs are better than total HSCs or Lin-HSCs, then why have statistical comparisons been made with PD-1 alone group in Fig 2?

5.Supplement Fig7A:

a.How were adoptively transferred T cells distinguished from tumor-bearing host T cells?

b.IFN γ (YFP) gate is not the same in BMDP2+ population

c.image for GFP+CCR2- HSC population is missing?

6.Supplement Fig 5: in the figure legend, it does not mention that PD-1 treatment was administered for part A. The figure legend mentions that T cells were harvested from these mice for part B where it mentions tumor+PD-1, HSC+PD-1, CCR-HSC+PD-1??

Reviewer #3 (Remarks to the Author):

Manuscript#: NCOMMS-17-24581

Title: Lin-CCR2+ hematopoietic stem cells overcome resistance to PD-1 blockade in brain tumors

Summary: The authors present an elegant study demonstrating for the first time that incorporation of CCR2+ Hematopoietic Stem Cell (HSC) rescue sensitizes malignant brain tumors to PD-1 blockade treatment as well as to adoptive T cell therapy. The authors present a study in multiple tumor models such as glioblastoma, diffuse intrinsic pontine gliomas and medulloblastoma. As expected, the antitumor efficacy of the treatment was associated with increased T cell responses. Elicitation of this immune response was dependent on CCR2+ HSCs instead of their CCR2- HSCs counterpart. These CCR2+ HSCs were capable of migrating into brain

tumor deposits, differentiate into antigen presenting cells and activate tumor specific T cell responses when combined with PD-1 blockade treatment and adoptive T cell therapy. These results demonstrate the potency and generalization of this novel approach to multiple brain tumor with different etiology, and overcomes a major hurdle in the field of brain tumors by sensitizing tumor to PD-1 blockade, and enhance the efficacy of adoptive therapy.

Comments to the author

Major comments

Minor comments

1. Please make sure that the size of the text is all the same throughout the manuscript.
2. Please indicate and provide flow plots demonstrating the purity of CCR2+ and CCR2- HSC enrichment procedure in supplementary data.
3. While Figures 1 and 2 demonstrate the elicitation of endogenous T cell responses upon CCR2+ HSCs transfer, T cell depletion studies will further demonstrate if these are responsible for the antitumor efficacy.
4. Inclusion of MHC I^{-/-} or MHC II^{-/-} CCR2+ or CCR2- HSC (as presented in supplemental figure 9) in experiments incorporating HSCs transfer + PD-1 blockade, but not adoptive t cell transfer, would have strengthened the conclusion crosspresentation by the transferred HSCs in the context of checkpoint blockade.
5. The data presented in Supplementary figures 7, 10 & 11 are elegant studies performed in the context of adoptive T cell therapy. Have these being performed in the context of HSCs transfer + PD-1 blockade treatment presented in figure 1 & 2?
6. The figure legend for Supplementary figure 7B does not match the description in the text of figure. However, do these different HSC subsets presented in Supplemental Figure 7B have an impact of sensitizing brain tumors to check point blockade or is it just the CCR2+ as well.

We thank the reviewers for their insight into the project and data presented in this manuscript. The revisions incorporating suggestions and addressing questions from the prior review have significantly improved our manuscript. We have provided line item responses in the following pages to each comment that was raised in the review. We appreciate your contributions that we believe have significantly enhanced the breadth and depth of our investigation. We hope that the reviewers find this to be a significantly improved report.

Reviewer #1

This manuscript focuses on the ability of hematopoietic stem cells (HSCs) to enhance immune responses to brain tumors. Using syngeneic mouse models of glioblastoma (GBM) and medulloblastoma (MB), the authors demonstrate that HSCs expressing CCR2 can enhance the effects of PD-1 blockade or adoptive T cell therapy. They show that intravenously administered CCR2+HSCs can migrate to intracranial tumors and differentiate into CD11c+ antigen presenting cells (APCs) at the tumor site. These APCs can present tumor-derived antigens to CD4+ and CD8+ T cells, leading to T cell activation and enhanced tumor rejection. Based on these studies, they conclude that CCR2+HSCs could be used to overcome resistance to PD-1 blockade and adoptive cellular therapy.

The main concern of this reviewer was the presentation of the data and the way the manuscript was previously written did not convey the importance of our findings. We thank you for this assessment and we have completely re-write the manuscript into a full article instead of letter or short report which is much more thorough and understandable.

1. “Overall, this is an extremely interesting paper that has important implications for therapy of brain tumors. The experiments are generally well designed and controlled, and the conclusions are supported by the data. My main concern is that the paper is very difficult to follow.” The manuscript has been rewritten and reformatted to address this concern.
2. There are many places where figures are not properly labeled, terms and abbreviations are not well defined, and experimental methods are not described in sufficient detail. The manuscript has been rewritten and reformatted to address this concern.
3. In addition, a large proportion of the important data is relegated to supplemental figures, which is not a problem for reviewers but will make the paper difficult to read if and when it is published. The manuscript has been rewritten and reformatted to address this concern.
4. These issues are prevalent enough that they interfere with interpretation of the results and appreciation of their significance. While very few new experiments are necessary, a cleaned-up, simplified, user-friendly version of the manuscript should be provided. The manuscript has been rewritten and reformatted to address this concern.
5. The Abstract and Introduction seem redundant. Since the Abstract summarizes the key findings of the paper, it would be helpful if the Introduction was used to present relevant background that is necessary to understand the significance of these studies The manuscript has been rewritten and reformatted to address this concern.
6. The manuscript contains virtually no Discussion section. Notwithstanding the word limits, it would be valuable to include at least a paragraph or two discussing the implications of this work for the field. The manuscript has been rewritten and reformatted to address this concern.
7. One question that should be addressed – either experimentally or through discussion – is why endogenous HSC's don't migrate to the CNS tumor microenvironment and mediate the same effects as the transplanted HSCs in these studies. The migration of endogenous bone marrow derived HSCs to intracranial gliomas has been previously described through the CXCR4/SDF-1 axis¹⁻³. Since it is well characterized that endogenous bone marrow derived stem and progenitor cells migrate to intracranial tumor, and it is known that gliomas are infiltrated by MDSCs and other suppressive myeloid cells (i.e. tumor associated macrophages), we postulate that the endogenous HSCs are likely skewed to differentiate into suppressor cells. In the context of combination with immunotherapy, we believe these HSCs preferentially differentiate into potent antigen presenting cells as demonstrated in this manuscript. Additionally, there may be differences in the homeostatic mobilization of HSCs

into the periphery versus harvesting and intravenous injection that leads to quantitative and qualitative differences. We have addressed these considerations in the modified discussion and is the subject of ongoing experimentation within our laboratory.

8. A large proportion of the data is presented in Supplementary Figures. Since much of this information is critical to the logic of the paper, I would suggest incorporating some of the data back into the main figures. The manuscript has been rewritten and reformatted to address this concern.

9. Like the main figure legends, the supplemental figure legends should have titles that highlight the point of the figure, rather than just launching into the details of the experiment. The manuscript has been rewritten and reformatted to address this concern.

10. On p. 4, the authors refer to patched tumors as representing Group 3 medulloblastoma. This is incorrect; they represent sonic hedgehog-driven medulloblastoma. Thank you for catching this. We have corrected.

11. In many of the figures, the authors refer to experimental groups as “tumor only”, “HSC only,” “Anti-PD-1 only,” etc. This is confusing: they should either refer to the tumor-only group as “no treatment”, or refer to the treatment groups as “tumor + HSCs,” “tumor + Anti-PD-1”, etc. The manuscript has been rewritten and reformatted to address this concern.

12. Many of the FACS plots (e.g. Supplemental Figures 4, 5 and 8) are very difficult to read. The Y-axes are illegible, and there are no numbers shown to indicate the percentages of cells in the red boxes. Without this information, the reader cannot assess the results properly. The manuscript has been rewritten and reformatted to address this concern.

13. The authors should indicate how many replicates they did for each and every experiment in the paper. Statements like “this has been repeated in subsequent experiments” are not sufficient The manuscript has been rewritten and reformatted to address this concern.

14. The authors do not provide enough detail regarding their method for purifying HSCs; they only mention this in the methods, referring to a kit from Miltenyi. Since HSCs are the focus of the entire manuscript, it would be helpful to explain this in detail. The HSCs used in this study were isolated from fresh bone marrow using a magnetic bead lineage depletion kit (Miltenyi Biotec). The resulting lineage negative population is depleted of CD5, CD45R, CD11b, Gr-1, 7-4, and Ter-119. This experimental detail has been added to the manuscript.

15. In Figure 1A, it is noteworthy that about half the mice treated with HSC + Anti-PD-1 die with kinetics similar to the other treatment groups, and the other half show prolonged survival. This bimodal effect is interesting, and suggests that only some mice are responsive to HSCs. If this is a reproducible observation, the authors should comment on it. This observation is reproducible as shown in the figures in two preclinical models of brain tumors, and has been observed in at least three additional experiments. Although we have unveiled one mechanism by which HSC + PD-1 increases efficacy against brain tumors, this bimodal effect suggests that there may be other mechanisms that play a role in the observed efficacy. Finding a biomarker to predict response to therapy would be an impactful finding. We have addressed this observation in the discussion of the manuscript.

16. In Figure 2A, the effect of unsorted HSCs + anti-PD-1 is much weaker than in Figure 1A. Which is a more representative result? Figure 2A is the more representative result and has been repeated in three additional experiments. We have conducted the experiments and have empirically determined that this difference is due to the sensitivity of the tumors to different anti-PD-1 monoclonal antibody clones (Merck DX400, Bxcell J43, Bxcell RMP1-14, Bxcell J116; shown in figure below). Although there is a difference in median survival, all clones lead to the same proportion of long term survivors. We have since chosen a single clone, and have repeated all experiments looking at T cell activation with the same clone. All mice received 10mg/kg of anti-mouse PD-1 every five days for a total of four administrations regardless of the monoclonal antibody clone.

17. In Supplemental Figure 3, it is not clear what the numbers across the top and the letters on the left side represent. Do different rows (or columns) represent different mice? Or is the whole heat map a representation of the ratio of expression in one group of mice versus the other? This needs to be clarified in the legend. And rather than simply pointing to a few of the spots on the array, the identity of all the spots should be shown in a table or list accompanying the figure. This figure has been inserted into Figure 3 alongside a plate map describing what all genes were analyzed along with a more thorough description of the experiment. Here, Tumor bearing mice were treated with either CCR2⁺HSC + PD-1 or CCR2⁻HSC + PD-1. Tumors were excised and RNA was isolated to determine relative gene expression of tumors isolated from mice that received CCR2⁺HSCs relative to those that received CCR2⁻HSCs. Genes analyzed were in the RT2PCR Array for T- and B- Cell Activation (Qiagen). Genes represented demonstrated >2 fold change in expression.

18. For the experiment in Supp Figure 4, based on the x-axis label, the authors seem to use dsRed to identify donor-derived cells. This is a critical element of the experiment, but it is not mentioned in the text or the legend. The details of this experiment should be clarified. The manuscript has been rewritten and reformatted to address this concern.

19. For Supp Figure 5, was the phenotype of the dsRed⁺ cells in the lymph nodes determined? Do the authors know if these cells included dendritic cells, macrophages, or even T cells? This information is important for interpreting the results of this experiment. The manuscript has been rewritten and reformatted to address this concern.

20. The PCR array in Supp. Figure 6 has the same issues as the one in Supp. Figure 3. In addition, the significance of the arrow pointing to “IFN?” is unclear. Are they not sure that this spot represents interferon? Or is the “?” supposed to be a gamma? This needs to be corrected. And all the probes on the array should be identified, rather than just pointing out the one that they are interested in. This was removed and addressed as described in response to critique #17.

21. On p. 10, discussing Supp. Figure 7C, the authors say “To determine if these HSC subsets migrate to intracranial tumor, DsRed⁺CCR2⁺HSCs and GFP⁺CCR2⁻HSCs were intravenously injected in equal amounts (5x10⁵ cells) into lymphodepleted tumor-bearing mice.” How is this experiment different from the previous experiments involving dsRed-expressing HSCs (e.g. Supplemental Figure 4)? Haven’t they already established that the cells can migrate to the brain? If the experiments are redundant, one should be removed. If they are complementary, this should be clarified, and the authors should consider combining these data into one figure. The purpose of the experiment in Supplemental Figure 7 was to compare preferential migration of CCR2⁺ versus CCR2⁻HSCs into brain tumor. We had previously described that HSCs migrate to tumor, but this experiment was to determine the difference in relative amounts of each CCR2⁺ versus CCR2⁻ population recruited to the brain tumor immediately after cell transfer. We found that a larger proportion of CCR2⁺ cells are recruited to tumor within 24hrs of cell transfer. We also show in **Figure 4A** (4A copied below) that there is a difference in the accumulation of cells derived from CCR2⁺HSCs and CCR2⁻HSCs over time. We found that there is more immediate (within 24hrs) migration of CCR2⁺HSCs to the brain, and that these cells upregulate a phenotype

consistent with antigen presenting cells **Figure 4B** (4B copied below). On the contrary, the cells derived from CCR2-HSCs that migrate to tumor, do so within 7 days of transfer and upregulate a phenotype consistent with monocyte derived suppressor cells as shown in **Figure 4D** (4D copied below). We believe this is an important distinction between the two cell populations and have clarified the importance of the experimental data in the manuscript.

22. In the text (pp. 10-11) describing Supp Fig 8, the authors state that “Antigen presenting cells from CCR2+HSCs had a distinct dendritic cell phenotype while cells that arose from CCR2-HSCs upregulated expression of monocyte suppressor cell marker Ly6G (Gr1-1)”. Is the dendritic cell phenotype determined only by IA-E? The legend to Supp Figure 8 mentions a number of other markers (CD11c, CD80, CD86), but the data for these markers are not presented. The authors need to provide stronger evidence for the DC vs. monocyte phenotype. We have conducted a more sufficient phenotype, describing that CCR2+HSCs differentiate into CD11c+, MHC-II+, CD80+, and CD86+ cells. Functionality assay also demonstrates that these cells can be pulsed with total tumor RNA and be used as targets for tumor-reactive T cells, resulting in IFN γ secretion, indicating recognition of cognate antigen, and assuming antigen presentation (**Figure 4E**). In this same assay, we demonstrate that the cells derived from CCR2-HSCs do not efficiently stimulate tumor-reactive T cells to physiologically relevant levels, indicating that these cells are not antigen presenting cells. Their upregulation of F4/80, Ly6G6C, and CCR2 indicate a suppressive phenotype (**Figure 4D and Supplemental Figure 1**).

23. In Supp Figure 9A there is no explanation for “Brain” and “Kluc cells” and no axis or label that explains the cells listed above the line (are these effectors?) and below the line (targets)? The manuscript has been rewritten and reformatted to address this concern.

24. It is not clear what the FACS plot in Supp Figure 9C is showing The manuscript has been rewritten and reformatted to address this concern.

25. The experiment depicted in Supp Fig 10B needs to be explained more clearly. The manuscript has been rewritten and reformatted to address this concern.

26. In Figure 3, there is no explanation for the term “MA” The manuscript has been rewritten and reformatted to address this concern.

27. The brain stem glioma cells used in Supp Figure 11 are not described in any detail. What mouse model are they from? This figure is deleted.

Reviewer 2

We appreciate that comments and insight of Reviewer 2. The questions raised were questions that were important in understanding the role of CCR2+HSC-derived cells. A more through comparison between CCR2+

and CCR2- HSC significantly emphasized our findings. We have found differences in functionality and phenotype between the cells that arise from the two populations, CCR2+HSCs being immune potentiating in the tumor microenvironment, and cells arising from CCR2-HSCs being immunosuppressive. In addition, the reviewer's suggestions to conduct fundamental questions regarding route of HSC administration, kinetics of migration of CCR2+ cells, and cell fate of CCR2+ versus CCR2-HSC derived cells were all conducted and further revealed biological distinctions between the two populations. All suggested changes have been made within the main text of the manuscript and are discussed below. Thank you.

The study titled “Lin-CCR2+ hematopoietic stem cells overcome resistance to PD-1 blockade in brain tumors” demonstrates that bone marrow-derived, lineage-negative hematopoietic stem and progenitor cells (HSCs) that express CCR2+ reverse treatment resistance and sensitizes mice to curative immunotherapy. Although authors have performed extensive survival based studies in mice bearing different intracranial tumor types with appropriate controls, fundamental questions with regards to:

1. Their delivery via different routes of administration. This was an excellent suggestion by the reviewer. We conducted an experiment to address this and have included it in the manuscript in **Figure 2A** (graph also included below). In these studies, HSCs have been administered intravenously since in the clinical setting blood and marrow stem and progenitor cell transfers are typically conducted via intravenous infusion. However, the biological activity we have been studying has been in the tumor bed, therefore we conducted experiments to determine if direct administration of HSCs into the tumor bed using stereotactic injection leads to increased efficacy. C57BL/6 mice were stereotactically implanted with KR258B glioma. Five days later, equal numbers of HSCs were administered either directly into the tumor bed via stereotactic injection, or via tail vein intravenous injection. Tumor bearing mice received either no treatment, PD-1 only, intracranial HSCs only, intravenous HSCs only, intracranial HSC + PD-1, or intravenous HSC + PD-1. Groups were then followed for survival. Administering HSCs directly into the tumor did not provide any survival benefit when combined with PD-1 relative to the no treatment group. The group that received intravenous HSCs + PD-1 had significant increase in median survival over the intracranial HSC + PD-1 group indicating that the route of administration of the cells is crucial to efficacy. These results suggest that biological activity that occurs between HSC infusion and the time it reached the brain tumor may strongly effect outcomes.

2. The rationale of combining the Lin- CCR2+ HSC with anti-PD1 therapy is not presented. Thank you for pointing this out! We have now clarified this in the body of the manuscript. Our prior publication demonstrated that the co-administration of HSCs with adoptive T cell therapy leads to increased recruitment and activation of the adoptively transferred tumor-reactive T cells within the tumor. Since PD-1 activity has been shown to increase T cell activation in cancer, we sought to determine if combining HSCs and PD-1 against refractory tumors would be efficacious. Since lineage negative HSCs is a heterogeneous population of stem and progenitor cells, we sought to identify the subpopulation responsible for the increased intratumor T cell population observed in our

studies. It has been previously characterized that monocyte progenitor cells require CCR2 expression to cross the blood-brain barrier into the CNS, and since the observed activity is in the brain, we included CCR2+HSCs in our analysis. As discussed in the body of the manuscript and **Figure 3A**, we examined the capacity of several stem and progenitor cell populations to maintain T cell activation within the brain tumor. We found that CCR2+HSCs led to increased T cell activation within the typically immunosuppressive glioma, thus we chose to combine CCR2+HSCs with PD-1 therapy.

3. Specifically, the authors claim enhanced recruitment of Lin-CCR2+ HSCs, however, detailed analysis of the HSC recruitment within the whole tumors has not been studied. Labelling HSC with luciferase should help following the recruitment of systemically delivered HSC. Additionally, whether or not these cells have better differentiation capabilities than the CCR2-Lin- HSCs needs to be more clearly established. Time kinetics experiments are needed to fully elucidate that CCR2+ HSCs are better than CCR2- HSCs. As observed by the authors, Lin-CCR2+ HSCs are not very efficient in rescuing hosts from lethal irradiation, these cells appear to be a 'more- developed' (and potentially developed) from their lin-CCR2- HSC counterparts (Fig. 3A). This is also supported by the finding that Lin-CCR2- HSCs become Lin-CCR2+ HSCs post-transfer (suppl Fig 4). It can be hypothesized that author's observations have a temporal effect: CCR2-Lin-HSCs become CCR2+ and mature to have better antigen presenting efficacy. We have conducted experiments to determine the kinetics of the recruitment of CCR2+ and CCR2- HSCs to the tumor and have included the results in **Figure 4A-4D** and **Supplementary Figure 1** (copied below). We found that within the first 24hs of cell administration, a larger proportion of CCR2+HSCs are found at the tumor relative to cells derived from CCR2-HSCs. However, by 7 days, we find larger proportions of cells derived from CCR2-HSCs are found at the tumor site, and this pattern remains at 30 days post-transfer. Interestingly, characterization of these cells show that cells derived from CCR2+HSC are consistent with an immune activating phenotype (MHC class II^{pos}, Ly6G/6C^{neg}, F4/80^{neg}) while cells derived from CCR2-HSCs are consistent with a regulatory or immunosuppressive phenotype (MHC class II^{pos}, Ly6G/6C^{pos}, F4/80^{pos}). We also demonstrate functional capacity of the cells derived from CCR2+HSCs to mediate efficient antigen presentation while CCR2- HSCs failed to functional as antigen presenting cells (**Figure 4E**).

Figure 4.

Supplementary Figure 1.

FIGURE 4A. Equal numbers of CCR2-HSCs (isolated from GFP+ mice) and CCR2+HSCs (isolated from DsRed mice) were injected into tumor bearing mice. Relative amounts of cells derived from either GFP+CCR2-HSCs and DsRed+CCR2+HSCs were compared at 1 day (*p=0.0010), 7 days (**p=0.0013), and 30 days (**p=0.0003) post-transfer (Mann-Whitney tests).

FIGURE 4B. In a separate experiment, CCR2+HSCs were isolated from GFP+ mice and intravenously co-transferred into tumor bearing mice that received adoptive cell therapy with tumor-reactive T cells. Twenty one days post-transfer, GFP+ cells found in the tumor were phenotyped for makers of dendritic cells.

FIGURE 4C/D. Both CCR2-HSCs and CCR2+HSCs were isolated from GFP+ mice and co-transferred into tumor bearing mice that received adoptive cell therapy with tumor-reactive T cells. Twenty one days post-transfer, GFP+ cells found in the tumors were analyzed for markers of a suppressive phenotype.

SUPPLEMENTAL FIGURE 1. KR158B glioma tumor bearing mice received adoptive transfer of equal numbers of GFP+CCR2-HSCs and DsRed+CCR2+HSCs. Tumors were excised and GFP+ and DsRed+ cells phenotyped for markers of MDSCs. (n=5/group, unpaired t-test).

Other concerns:

4. The abstract states “but these findings are restricted to occur only within intracranial brain tumors.” What is the rationale behind the claim of brain tumor specific therapeutic efficacy? I don’t see any other tumor models tested in this study. The text has been modified to address this concern. We have removed this statement.

5. Supplement Fig 4:

a. pre-transfer characterization of the population of HSCs before being transferred is essential (wt LY6C, MHC-II and CD11c) to demonstrate their antigen presenting potential.

b. Do not see DS-Red negative Ly6G/C+ve cells? This experiment was repeated and shows the appropriate flow plots.

c. Please comment on CCR2-ve cells have become CCR2+ve and vice-versa. It is important to note that one week after cell transfer, cells originally derived from CCR2+HSC downregulated CCR2 expression and have a very low frequency of either F4/80 or Ly6G/6C+ cells, indicating the lack of myeloid derived suppressor cells (MDSC) phenotype (Figure 4C). In contrast, cells derived from CCR2-HSCs upregulated markers associated with MDSCs including F4/80, Ly6G/6C and CCR2 (Figure 4D). To determine the relative amount of CCR2+HSC-derived cells versus CCR2-HSC-derived cells that upregulate a suppressive phenotype, we injected 5×10^5 of either DsRed+CCR2+HSC or GFP+CCR2-HSC into tumor bearing mice. Seven days post-transfer, brain tumors were harvested and HSC-derived cells were isolated and phenotyped for F4/80, Ly6G/6C, and CCR2 (Supplemental Figure 1). We found that relative to CCR2+HSC-derived cells, once in the brain tumor, cells that arose from CCR2-HSCs expressed significantly more CCR2 ($p=0.0035$), F4/80 ($p=0.0138$), and Ly6G/6C ($p=0.0001$)(Supplemental Figure 1). This is a strong indicator that both cell populations have different functions within the tumor.

3. Figure 1 shows 40-50% survival in anti-PD-1+HSC group while in Figure 2, similar survival is reported for PD-1 treatment+CCR2+HSC. For PD-1+HSC group, Survival is only 20% in Figure 2 experiment? There seems to be variation in data obtained from different experiments?

Figure 2A is the more representative result and has been repeated in three additional experiments. We have conducted the experiments and have empirically determined that this difference is due to the sensitivity of the tumors to different anti-PD-1 clones (Merck DX400, Bxcell J43, Bxcell RMP1-14, Bxcell J116; shown in figure below). Although there is a difference in median survival, all clones lead to the same proportion of long term survivors. We have since chosen a single clone, and have repeated all experiments looking at T cell activation with the same clone. All mice received 10mg/kg of anti-mouse PD-1 every five days for a total of four administrations regardless of the clone.

4. If the authors claim that CCR2+Lin-HSCs are better than total HSCs or Lin-HSCs, then why have statistical comparisons been made with PD-1 alone group in Fig 2?

We have added statistical testing comparing efficacy of CCR2+HSCs + PD-1 versus total HSCs + PD-1.

Figure 1

5. Supplement Fig7A:

a. How were adoptively transferred T cells distinguished from tumor-bearing host T cells?

In this experiment (which is now Figure 3D), wildtype C57BL/6 mice with established glioma received adoptive transfer of tumor-reactive T cells generated from GREAT mice (which have a YFP reporter on the IFN γ promoter). This was followed by administration of either CCR2+HSCs or CCR2-HSCs isolated from wildtype C57BL/6 mice. One week post-transfer, tumors were excised and analyzed by flowcytometry for CD3+YFP+ cells. Since only the tumor-reactive T cells were derived from GREAT mice, any YFP detected could only come from the adoptively transferred tumor-reactive T cells. We found that the cohort that received tumor-reactive T cells followed by CCR2+HSCs had a significantly higher proportion of T cells that expressed YFP within the tumor microenvironment.

b. IFN γ (YFP) gate is not the same in BMPR2+ population. This has been removed.

c. image for GFP+CCR2- HSC population is missing? The images have been removed.

6. Supplement Fig 5: in the figure legend, it does not mention that PD-1 treatment was administered for part A.

The figure legend mentions that T cells were harvested from these mice for part B where it mentions tumor+PD-1, HSC+PD-1, CCR-HSC+PD-1?? This has been corrected.

Reviewer #3

We thank Reviewer 3 for their comments. Although there were no major comments, the minor comments were actually very insightful and led us to conduct a few experiments to bolster our findings and answer some fundamental questions.

Major comments

There were no major comments

Minor comments

1. Please make sure that the size of the text is all the same throughout the manuscript. This has been addressed.

2. Please indicate and provide flow plots demonstrating the purity of CCR2+ and CCR2- HSC enrichment procedure in supplementary data.

3. While Figures 1 and 2 demonstrate the elicitation of endogenous T cell responses upon CCR2+ HSCs transfer, T cell depletion studies will further demonstrate if these are responsible for the antitumor efficacy.

We conducted T cell depletion experiments to determine if CD4 or CD8 T cells are major contributors to efficacy of this platform.

4. Inclusion of MHC1-/- or MHCII-/- CCR2+ or CCR2- HSC (as presented in supplemental figure 9) in experiments incorporating HSCs transfer + PD-1 blockade, but not adoptive t cell transfer, would have strengthened the conclusion crosspresentation by the transferred HSCs in the context of checkpoint blockade. We conducted experiments to confirm that CCR2+HSC-derived cells cross-present antigen via MHC-I during combinatorial therapy with PD-1. To do this, GREAT mice were implanted with intracranial gliomas then treated with PD-1. We

then isolated CCR2⁺ and CCR2⁻ HSCs from bone marrow of wildtype C57BL/6 mice, MHC-I^{-/-} mice, and MHC-II^{-/-} mice. GREAT mice bearing late stage intracranial tumors were then treated with PD-1 and either no treatment, PD-1 only, CCR2⁺HSC + PD-1, CCR2⁺MHC-I^{-/-} HSC + PD-1, CCR2⁺MHC-II^{-/-} HSC + PD-1, CCR2⁻HSC + PD-1, CCR2⁻MHC-I^{-/-} HSC + PD-1, or CCR2⁻MHC-II^{-/-} HSC + PD-1 (**Figure 8A**). One week after treatment, tumors were excised and processed into single cell suspensions. Flowcytometric analysis was conducted to determine expression of YFP on CD3⁺ cells. Results demonstrate significantly more YFP⁺CD3⁺ cells in mice that received CCR2⁺HSC + PD-1 (mean YFP⁺CD3⁺ cells = 14.41% ±1.729, n=5) than PD-1 alone (mean YFP⁺CD3⁺ cells = 3.58 ± 0.945, n=5; p=0.0006)(**Figure 8A**). The relative amount of YFP⁺CD3⁺ cells in tumor was also significantly decreased in the group that received CCR2⁺MHC-I^{-/-} HSC + PD-1 (mean YFP⁺CD3⁺ cells = 0.597% ± 0.2487, n=5) relative to the group that received CCR2⁺HSC + PD-1 (p=<0.0001). Interestingly, we also observe a significant decrease in YFP⁺CD3⁺ cells in the group that received CCR2⁺MHC-II^{-/-} HSC + PD-1 (1.196% ± 0.5061, n=5, p=0.0001), suggesting at the least, that presentation to both CD4 and CD8 T cells are necessary for the observed intratumor T cell activation. We also observed no significant increase in YFP⁺CD3⁺ within the tumor in any groups that received CCR2⁻HSCs relative to the no treatment group. This further confirms our observations that CCR2⁺HSCs are responsible for increased activation of tumor infiltrating T cells.

5. The data presented in Supplementary figures 7 (Fig 1C/D), 10 & 11 (Fig 8B/c) are elegant studies performed in the context of adoptive T cell therapy. Have these being performed in the context of HSCs transfer + PD-1 blockade treatment presented in figure 1 & 2? **Excellent question.** These studies were conducted with adoptively transferred T cells in order to allow longitudinal tracking of T cell fate and activation. We then sought to determine the impact of HSC transfer on T cells with PD-1. To determine the effect of HSC co-transfer on the relative amounts of activated tumor infiltrating lymphocytes, we employed GREAT mice which have an interferon-gamma (IFN γ) promoter with IRES-eYFP reporter (Jackson laboratories) to enable longitudinal evaluation of T cell activation within the tumor microenvironment. Here, GREAT mice were implanted intracranially with KR158B glioma in the cerebral cortex. These tumor-bearing mice then received either no treatment, HSCs alone, PD-1 alone, or combination HSC + PD-1. Three weeks after HSC transfer, tumors and draining cervical lymph nodes were excised and analyzed for YFP⁺ expression by CD3⁺ cells as an indication of activation and IFN- γ secretion of T cells in the tumor microenvironment (**Figure 1C**) and tumor draining lymph nodes (**Figure 1D**). Mice that received combinatorial HSC + PD-1 had significantly increased YFP⁺CD3⁺ cells within the tumor (p=0.0079) (**Figure 1C**) and draining lymph nodes (**Figure 1D**) relative to groups that received PD-1 alone (p=0.0079). When we further fractionate the HSC population into CCR2⁺ and CCR2⁻HSCs, and administer those with PD-1, we find superior survival against both KR158B glioma and Ptc medulloblastoma (**Figure 8B/C**).

Figure 1

Figure 8

Figure 1C/D: C) GREAT mice with established KR158B tumors received either no treatment, HSCs only, PD-1 only, or HSC + PD-1. Tumors were excised and analyzed for relative expression of YFP using flowcytometry to compare relative YFP+CD3+ expression in mice that received HSC + PD-1 only relative to PD-1 only (* $p=0.0079$, Mann-Whitney test; $n=5$ mice/group). D) GREAT mice with established KR158B tumors received either no treatment, HSCs only, PD-1 only, or HSC + PD-1. Tumor draining lymph nodes were excised and analyzed for relative expression of YFP using flowcytometry to compare relative YFP+CD3+ expression in mice that received HSC + PD-1 only relative to PD-1 only (* $p=0.0079$, Mann-Whitney test; $n=5$ mice/group).

Figure 8B/C: B) KR258B tumor bearing mice received no treatment, HSC only, PD-1 only, CCR2⁻HSCs, CCR2⁺HSCs, HSCs + PD-1, CCR2⁻HSCs + PD-1, or CCR2⁺HSCs + PD-1 (* $p=0.0233$, ** $p=0.0006$). C) Ptc medulloblastoma tumor bearing mice received no treatment, HSC only, PD-1 only, CCR2⁻HSCs, CCR2⁺HSCs, HSCs + PD-1, CCR2⁻HSCs + PD-1, or CCR2⁺HSCs + PD-1 (* $p=0.0001$, ** $p=0.0005$).

6. The figure legend for Supplementary figure 7B does not match the description in the text of figure. However, do these different HSC subsets presented in Supplemental Figure 7B have an impact of sensitizing brain tumors

to check point blockade or is it just the CCR2+ as well. We have examined these stem and progenitor cell populations only on their capacity to maintain intratumor T cell activation. Although very interesting, we have not yet determined if there are other mechanisms by which these progenitor cell populations might sensitize the tumor to checkpoint blockade. This is a focus of ongoing work in our laboratory.

- 1 Flores, C. *et al.* Novel role of hematopoietic stem cells in immunologic rejection of malignant gliomas. *Oncoimmunology* **4**, e994374, doi:10.4161/2162402X.2014.994374 (2015).
- 2 Breznik, B., Motaln, H., Vittori, M., Rotter, A. & Lah Turnsek, T. Mesenchymal stem cells differentially affect the invasion of distinct glioblastoma cell lines. *Oncotarget* **8**, 25482-25499, doi:10.18632/oncotarget.16041 (2017).
- 3 Bryukhovetskiy, I. S. *et al.* Hematopoietic stem cells as a tool for the treatment of glioblastoma multiforme. *Mol Med Rep* **14**, 4511-4520, doi:10.3892/mmr.2016.5852 (2016).

REVIEWERS' COMMENTS:

Reviewer #1 (Remarks to the Author):

The authors have done an excellent job revising the manuscript. It is much clearer and more coherent, and the experiments and interpretation are much easier to follow. I believe that these studies represent a significant advance, and will be of interest to a broad range of cancer biologists and immunologists. I have just a few minor comments and suggestions:

1) The abstract launches immediately into the results, with no information about the question being asked or the rationale for the study. One or two sentences setting up the question would help the reader understand the significance of this work.

2) Since the authors ultimately demonstrate that the Lin-CCR2+ cells they are studying are not multipotent hematopoietic stem cells, but myeloid progenitors, it would be better to refer to them as hematopoietic progenitor cells, rather than hematopoietic stem cells/HSCs, throughout the paper. This includes the title, since the use of the term HSCs there may be particularly misleading.

3) On p. 11, referring to Supplemental Figure 2, the authors state that "Cells originally derived from the CCR2+ HSCs expressed a distinct dendritic cell phenotype including CD11C, MHC-II, CD80 and CD86." This implies that the phenotype of the cells derived from CCR2+ cells resembles that of dendritic cells, whereas that of the cells derived from CCR2- cells does not. But looking at the figure, cells derived from CCR2+ and CCR2- cells have remarkably similar phenotypes (with the possible exception of Gr-1, which is higher on the latter). The authors should clarify this in the text. This does not diminish their conclusion that these cell types are different, since they go on to show that the cells derived from CCR2+ cells are more effective at antigen presentation.

Reviewer #2 (Remarks to the Author):

The authors have performed new experiments and obtained data which addresses my main concerns on the route of delivery of HSCs, detailed analysis on the recruitment of systemically delivered HSC and the rationale of the combined therapies performed in this study.

Although, the manuscript still needs some work to correct the grammar and the flow of writing but it should be acceptable for publication.

Reviewer #3 (Remarks to the Author):

Manuscript #: NCOMMS-17-24581A

Manuscript title: Lin-CCR2+ hematopoietic stem cells overcome resistance to PD-1 blockade in brain tumors

The current re-submission has addressed all prior concerns from this reviewer. The article is concise, clear and utilizes state of the art models to address fundamental questions regarding the mechanisms by which CCR2+HSCs overcome resistance to PD-1 blockade in brain tumors. Conclusions are supported by the experimental results and demonstrated in multiple models. The results are extremely impactful and promising for the field of brain tumor immunotherapy.

Major Comments

None

Minor Comments

None

Response to reviewer comments:

We thank the reviewers for their review and insight. The comments and suggestions were very helpful and resulted in a more comprehensive and thorough manuscript. We have addressed the comments from Reviewers 1 and 2 within the main manuscript.

Reviewer #1 (Remarks to the Author):

The authors have done an excellent job revising the manuscript. It is much clearer and more coherent, and the experiments and interpretation are much easier to follow. I believe that these studies represent a significant advance, and will be of interest to a broad range of cancer biologists and immunologists. I have just a few minor comments and suggestions:

1) The abstract launches immediately into the results, with no information about the question being asked or the rationale for the study. One or two sentences setting up the question would help the reader understand the significance of this work. **We have edited the abstract to address this.**

Immune checkpoint blockade using anti-PD-1 monoclonal antibodies has shown considerable promise in the treatment of solid tumors, but brain tumors remain notoriously refractory to treatment. In CNS malignancies that are completely resistant to PD-1 blockade, we found that bone marrow-derived, lineage-negative hematopoietic stem and progenitor cells (HSCs) that express C-C chemokine receptor type 2 (CCR2⁺) reverses treatment resistance and sensitizes mice to curative immunotherapy. HSC transfer with PD-1 blockade increases T cell frequency and activation within tumors in preclinical models of glioblastoma and medulloblastoma. CCR2⁺HSCs preferentially migrate to intracranial brain tumors and differentiate into antigen presenting cells within the tumor microenvironment and cross-present tumor-derived antigens to CD8⁺ T cells. HSC transfer also rescues tumor resistance to adoptive cellular therapy in medulloblastoma and glioblastoma. Our studies demonstrate a novel role for CCR2⁺HSCs in overcoming brain tumor resistance to PD-1 checkpoint blockade and adoptive cellular therapy in multiple invasive brain tumor models.

2) Since the authors ultimately demonstrate that the Lin⁻CCR2⁺ cells they are studying are not multipotent hematopoietic stem cells, but myeloid progenitors, it would be better to refer to them as hematopoietic progenitor cells, rather than hematopoietic stem cells/HSCs, throughout the paper. This includes the title, since the use of the term HSCs there may be particularly misleading.

Thank you for this. This is an important point. We have changed the title to: Lin-CCR2⁺ hematopoietic stem and progenitor cells overcome resistance to PD-1 blockade. We have also more clearly defined the use of HSC in the introduction.

3) On p. 11, referring to Supplemental Figure 2, the authors state that “Cells originally derived from the CCR2⁺ HSCs expressed a distinct dendritic cell phenotype including CD11C, MHC-II, CD80 and CD86.” This implies that the phenotype of the cells derived from CCR2⁺ cells resembles that of dendritic cells, whereas that of the cells derived from CCR2⁻ cells does not. But looking at the figure, cells derived from CCR2⁺ and CCR2⁻ cells have remarkably similar phenotypes (with the possible exception of Gr-1, which is higher on the latter). The authors should clarify this in the text. This does not diminish their conclusion

that these cell types are different, since they go on to show that the cells derived from CCR2+ cells are more effective at antigen presentation.

We have added the following sentence immediately after the previous statement: Cells derived from CCR2⁺HSCs express a similar phenotype but have notably high expression of Gr-1, indicating a suppressive phenotype (**Supplementary Fig. 2**).

Reviewer #2 (Remarks to the Author):

The authors have performed new experiments and obtained data which addresses my main concerns on the route of delivery of HSCs, detailed analysis on the recruitment of systemically delivered HSC and the rationale of the combined therapies performed in this study.

Although, the manuscript still needs some work to correct the grammar and the flow of writing but it should be acceptable for publication. We have edited the grammar and flow of the writing throughout the manuscript.

Reviewer #3 (Remarks to the Author):

Manuscript #: NCOMMS-17-24581A

Manuscript title: Lin-CCR2+ hematopoietic stem cells overcome resistance to PD-1 blockade in brain tumors

The current re-submission has addressed all prior concerns from this reviewer. The article is concise, clear and utilizes state of the art models to address fundamental questions regarding the mechanisms by which CCR2+HSCs overcome resistance to PD-1 blockade in brain tumors. Conclusions are supported by the experimental results and demonstrated in multiple models. The results are extremely impactful and promising for the field of brain tumor immunotherapy.

Major Comments

None

Minor Comments

None